# Metabolic regulation of misfolded protein import into mitochondria

Yuhao Wang[1,2†], Linhao Ruan[1†], Jin Zhu[3], Xi Zhang[1],
Alexander Chih-Chieh Chang[1,4], Alexis Tomaszewski[1,2], Rong Li[1,3,4*]

[1]Center for Cell Dynamics and Department of Cell Biology, Johns Hopkins University School of Medicine, Baltimore, United States; [2]Biochemistry, Cellular and Molecular Biology (BCMB) Graduate Program, Johns Hopkins University School of Medicine, Baltimore, United States; [3]Mechanobiology Institute and Department of Biological Sciences, National University of Singapore, Singapore, Singapore; [4]Department of Chemical and Biomolecular Engineering, Whiting School of Engineering, Johns Hopkins University, Baltimore, United States

*For correspondence:
rong@jhu.edu

†These authors contributed equally to this work

Competing interest: The authors declare that no competing interests exist.

**Abstract** Mitochondria are the cellular energy hub and central target of metabolic regulation. Mitochondria also facilitate proteostasis through pathways such as the 'mitochondria as guardian in cytosol' (MAGIC) whereby cytosolic misfolded proteins (MPs) are imported into and degraded inside mitochondria. In this study, a genome-wide screen in *Saccharomyces cerevisiae* uncovered that Snf1, the yeast AMP-activated protein kinase (AMPK), inhibits the import of MPs into mitochondria while promoting mitochondrial biogenesis under glucose starvation. We show that this inhibition requires a downstream transcription factor regulating mitochondrial gene expression and is likely to be conferred through substrate competition and mitochondrial import channel selectivity. We further show that Snf1/AMPK activation protects mitochondrial fitness in yeast and human cells under stress induced by MPs such as those associated with neurodegenerative diseases.

## eLife assessment

This study makes a connection between cellular metabolism and proteostasis through MAGIC, a previously proposed protein quality control pathway of clearance of cytosolic misfolded and aggregated proteins by importing into mitochondria. The authors reveal the role of Snf1, a yeast AMPK, in preventing the import of misfolded proteins to mitochondria for MAGIC controlled by the transcription factor Hap4, depending on the cellular metabolic status. The key message is **important**, although the evidence for physiological relevance of MAGIC for overall cellular proteostasis and its molecular regulation by Snf1 remains **incomplete**.

## Introduction

Mitochondria are vital organelles whose biogenesis and activities in energy production are tightly linked to cellular metabolic control (*Andréasson et al., 2019*; *Wai and Langer, 2016*). Metabolic stress and mitochondrial dysfunction are common drivers of age-related degenerative diseases such as heart failure and dementia (*López-Otín et al., 2023*; *Nunnari and Suomalainen, 2012*), which are often characterized by loss of proteostasis leading to the formation of protein aggregates (*López-Otín et al., 2023*; *Hipp et al., 2019*). In yeast, acute proteotoxic stress such as heat shock induces reversible protein aggregation in cytosol (*Zhou et al., 2011*; *Escusa-Toret et al., 2013*; *Zhou et al., 2014*; *Wallace et al., 2015*; *Ruan et al., 2017*). Protein aggregates are initially formed on the cytosolic surface of the endoplasmic reticulum, and later captured at the mitochondrial outer membrane

(*Escusa-Toret et al., 2013*; *Zhou et al., 2014*). Upon reversal to the stress-free condition, aggregates undergo dissolution that is not only dependent on the activity of the Hsp104 chaperone but also mito-chondrial membrane potential (MMP) (*Zhou et al., 2014*; *Ruan et al., 2017*). This observation led to a hypothesis that mitochondria play an active role in the clearance of cytosolic misfolded proteins (MPs). Using both imaging-based and biochemical assays, we showed that certain aggregation-prone native cytosolic proteins and the model aggregation protein firefly luciferase single mutant (FlucSM) (*Gupta et al., 2011*), but not stable cytosolic proteins, are imported into the mitochondrial matrix (*Ruan et al., 2017*). A subset of highly aggregation-prone proteins known as super-aggregators (*Wallace et al., 2015*) are imported into mitochondria even in the absence of heat stress (*Ruan et al., 2017*). Mitochondrial proteases, most prominently the LON protease Pim1, degrade the imported MPs in the mitochondrial matrix, and this pathway of clearance of cytosolic MPs was termed 'mitochondria as guardian in cytosol' (MAGIC) (*Ruan et al., 2017*; *Figure 1—figure supplement 1A*).

Cytosolic MPs have also been found in human mitochondria. Both FlucSM and a more destabilized double mutant (FlucDM) (*Gupta et al., 2011*), but not the well-folded wild-type Fluc (FlucWT) or gluta-thione *S*-transferase (GST), are imported into the mitochondrial matrix of human RPE-1 cells (*Ruan et al., 2017*). In HeLa cells, proteasomal inhibition by MG132 induces the mitochondrial import of unfolded cytosolic model protein in a manner dependent on mitochondrial outer membrane protein FUNDC1 and cytosolic chaperone HSC70 (*Li et al., 2019b*). Furthermore, disease-related proteins such as α-synuclein (αSyn), FUS, and TDP-43 are found in the mitochondria of human cells (*Devi et al., 2008*; *Deng et al., 2018*; *Wang et al., 2016*). These results suggest that a MAGIC-like pathway may exist in higher organisms, although the underlying mechanisms could be different.

It remains unclear whether MAGIC is beneficial or detrimental to cellular or mitochondrial fitness. Nevertheless, the MAGIC pathway may represent a link between mitochondrial dysfunction and loss of proteostasis. Although inhibition of mitochondrial import after heat shock causes prolonged protein aggregation in cytosol, an elevated burden of MPs in mitochondria can also cause mitochon-drial damage (*Ruan et al., 2020*). Understanding how mitochondria balance functions in proteostasis and metabolism may provide key insights into the maintenance of cellular fitness under stress during aging. In this work, we conducted an unbiased imaging-based genetic screen in yeast to uncover cellular mechanisms that regulate MAGIC. We identified Snf1, the yeast AMP-activated protein kinase (AMPK), as a negative regulator of MAGIC through transcriptional upregulation of nuclear-encoded mitochondrial genes. We also showed that AMPK activation in yeast and human cells attenuates mitochondrial accumulation of disease-related MPs and may protect cellular fitness under proteotoxic stresses.

## Results

### A genetic screening for regulators of MAGIC

To observe the mitochondrial import of cytosolic MPs, we employed a previously established method using split-GFP (spGFP) system in which the first 10 β-strand of GFP (GFP$_{1-10}$) was targeted into the mitochondrial matrix while the eleventh β-strand (GFP$_{11}$) was tagged with MPs (*Ruan et al., 2017*; *Figure 1—figure supplement 1B*). Because mitochondrial import requires substrate in an unfolded state (*Wiedemann and Pfanner, 2017*), globular GFP reconstituted in the cytosol should not be imported. Indeed, mitochondrial spGFP signal of stable cytosolic protein Hsp104 failed to increase after heat shock (*Ruan et al., 2017*; *Figure 1—figure supplement 1C*). In contrast, spGFP signals of FlucSM and several endogenous aggregation-prone proteins increased significantly after heat shock at 42°C compared to background at normal growth temperature (30°C) in WT cells (*Ruan et al., 2017*). Importantly, mitochondrial import of FlucSM and other misfolded cytosolic proteins after heat stress was further validated by using a variety of additional methods, including the classical biochem-ical fractionation and protease protection assay, APEX-based labeling in mitochondrial matrix, and super-resolution microscopy (*Ruan et al., 2017*).

To uncover cellular pathways that influence MAGIC, we performed a high-throughput spGFP-based genetic screen in the non-essential yeast knockout (YKO) collection (*Giaever et al., 2002*; *Figure 1A*). Briefly, for each mutant strain in this collection, Lsg1, one of the endogenous aggregation-prone proteins previously shown to be imported into mitochondria (*Ruan et al., 2017*), was C-termi-nally tagged with GFP$_{11}$ at *LSG1* genomic locus through homologous recombination. Also introduced

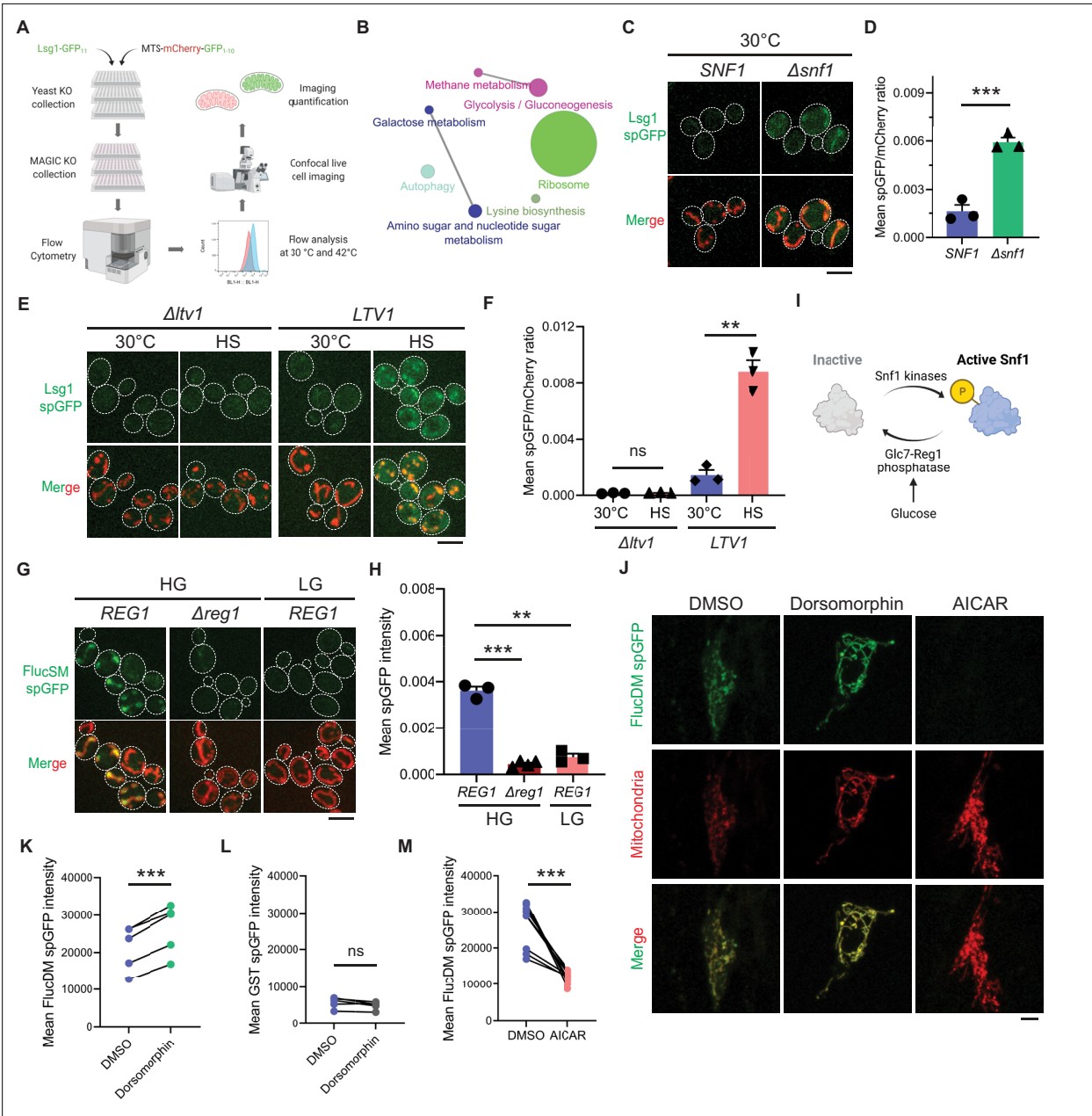

**Figure 1.** Mitochondria as guardian in cytosol (MAGIC) regulators revealed by a genome-wide screen in yeast and validations in human RPE-1 cells. (**A**) Workflow of the split-GFP (spGFP)-based genetic screen in yeast. (**B**) KEGG pathway analysis of validated mutants that affect MAGIC. The size of the node indicates the number of genes identified. Pathways with at least two associated genes are shown. (**C, D**) Representative images (**C**) and quantification (**D**) of Lsg1 spGFP signal in wild-type (WT) and *Δsnf1* cells at 30°C. Shown in (**C**): top, Lsg1 spGFP; bottom, merged images of spGFP and mitochondria labeled with MTS-mCherry. Shown in (**D**): means ± SEM of spGFP/mCherry ratio (n=3). Unpaired two-tailed *t*-test. (**E, F**) Representative images (**E**) and quantification (**F**) of Lsg1 spGFP signal in *Δltv1* and WT *LTV1* cells at 30°C and after HS. Shown in (**F**): means ± SEM of spGFP/mCherry ratio (n=3). Paired two-tailed *t*-test. HS: heat shock. (**G, H**) Representative images (**G**) and quantification (**H**) of FlucSM spGFP signals in WT (*REG1*) cells in HG or LG, and *Δreg1* cells in HG. Shown in (**G**): top, FlucSM spGFP; bottom, merged images of spGFP and mitochondria labeled with Tom70-mCherry. Shown in (**H**): means ± SEM of spGFP intensity (n=3 for *REG1*, n=4 for *Δreg1*). Paired (*REG1* in HG vs. LG) or unpaired (*REG1* vs. *Δreg1* in HG) two-tailed *t*-test. (**I**) Schematic diagram of Snf1 activation in yeast. (**J**) Representative images of FlucDM spGFP in RPE-1 cells treated with DMSO, dorsomorphin, or 5-aminoimidazole-4-carboxamide ribonucleoside (AICAR). Top, FlucDM spGFP; middle, mitochondria-targeted mCherry; bottom, merged images. (**K–M**) Flow cytometry-based quantifications of FlucDM spGFP in RPE-1 cells treated with DMSO, dorsomorphin, or AICAR (**K, M**), and glutathione *S*-transferase (GST) spGFP in cells treated with DMSO or dorsomorphin (**L**). Means ± SEM of spGFP intensities are shown. n=5 for (K) and (L). n=9 for (M). Paired two-tailed *t*-test. **p<0.01; ***p<0.001; ns, not significant, p>0.05. HG: 2% glucose; LG: 0.1% glucose plus 3% glycerol. Scale bars, 5 μm.

*Figure 1 continued on next page*

*Figure 1 continued*

The online version of this article includes the following source data and figure supplement(s) for figure 1:

**Source data 1.** Split-GFP (spGFP) quantification data.

**Figure supplement 1.** Schematics of mitochondria as guardian in cytosol (MAGIC) pathway and split-GFP (spGFP)-based imaging in the whole-genome screen in yeast.

**Figure supplement 1—source data 1.** Mean split-GFP (spGFP) intensity by flow cytometry.

**Figure supplement 2.** Snf1 regulates the accumulation of misfolded proteins in mitochondria after acute overexpression of FlucSM.

**Figure supplement 2—source data 1.** Raw data for split-GFP (spGFP) intensity and Mig1-GFP quantification.

**Figure supplement 2—source data 2.** Raw and labeled immunoblots for *Figure 1—figure supplement 2J*.

**Figure supplement 3.** Snf1 activation only modestly affects split-GFP (spGFP) reconstitution.

**Figure supplement 3—source data 1.** Quantification of split-GFP (spGFP) and mCherry intensity.

into each mutant strain was a construct constitutively expressing matrix targeted $GFP_{1-10}$ under the GAPDH promoter. $GFP_{1-10}$ was targeted into mitochondrial matrix by using the cleavable mitochondrial targeting sequence (MTS) of Subunit 9 of mitochondrial ATPase (Su9) from *Neurospora crassa*, and the red fluorescent protein mCherry was also included in this construct (MTS-mCherry-$GFP_{1-10}$), as previously described (*Ruan et al., 2017*). YKO mutants bearing the above Lsg1 spGFP reporter components were generated by using high-throughput transformation in 96-well plates. We used flow cytometry and analyzed Lsg1 spGFP signal of each mutant at 30°C and after 42°C heat shock for 30 min (*Figure 1—figure supplement 1D*). Mutants of interest were then subjected to hits validation using confocal fluorescence imaging. Based on mitochondrial spGFP intensity of each mutant and WT cells at two imaging time points, we classified the validated YKO mutants into two groups: five Class 1 mutants showed significant greater spGFP signal than WT at 30°C without heat shock, and 140 Class 2 mutants had no significant increase in spGFP signal after heat stress compared to 30°C (*Table 1*; details in Materials and methods).

KEGG pathway analysis revealed that genes corresponding to the hits validated with imaging encompassed many cellular pathways, most notably carbohydrate metabolism and ribosomal biogenesis (*Figure 1B*). Among five Class 1 mutants, a notable one is *Δsnf1* (*Figure 1C and D*; see further analyses below). Class 2 includes multiple genes related to ribosomal biogenesis (*Table 1*). For example, deletion of *LTV1* that encodes a chaperone required for the assembly of small ribosomal subunits (*Collins et al., 2018*) showed only baseline level Lsg1 spGFP fluorescence with no increase at 42°C (*Figure 1E and F*).

## Snf1/AMPK negatively regulates MP accumulation in mitochondria

In this study, we have chosen to focus on *SNF1*, as *SNF1* encodes the yeast homolog of the evolutionarily conserved AMPK which serves as a master nutrient sensor orchestrating the activation of glucose-repressed gene transcription and metabolic stress response in glucose-limited conditions (*Wright and Poyton, 1990*; *Hedbacker and Carlson, 2008*; *Hardie, 2007*). Its pivotal function in cellular metabolism and mitochondrial biogenesis spurred us to further examine its role in MAGIC. To avoid complicating effects of heat shock and to improve the sensitivity of spGFP reporter, we optimized our spGFP-based method to impose proteostasis burden by acute induction of the MAGIC substrate FlucSM (*Ruan et al., 2017*; *Gupta et al., 2011*) tagged with $GFP_{11}$ (FlucSM-$GFP_{11}$) via the β-estradiol-inducible system (*Costa et al., 2018*). $GFP_{1-10}$ was stably targeted to the mitochondrial matrix by fusion with a matrix protein Grx5 (Grx5-$GFP_{1-10}$). After induction upon β-estradiol treatment at 30°C for 90 min, FlucSM spGFP signal increased significantly within mitochondria compared to the ethanol-treated control (*Figure 1G and H*; *Figure 1—figure supplement 2A–C*; *Video 1*). The spGFP signal in mitochondria showed an increasing trend that positively correlated with the structural instability of luciferase-derived MPs: FlucWT, FlucSM, and FlucDM with the highest structural instability (*Gupta et al., 2011*; *Figure 1—figure supplement 2D and E*). We chose to use the intermediate construct, FlucSM-$GFP_{11}$, for testing the effects of modulating Snf1 activity on mitochondrial import of MPs.

Reg1 is the regulatory subunit of Glc7-Reg1 protein phosphatase 1 complex that dephosphorylates Snf1 and promotes its inhibitory conformation (*Tu and Carlson, 1995*; *Ludin et al., 1998*; *Sanz et al., 2000*; *Ruiz et al., 2011*). Either glucose limitation or loss of Reg1 in glucose-rich medium (HG:

**Table 1.** List of validated mitochondria as guardian in cytosol (MAGIC) regulators.
Bold: ribosome-associated genes based on KEGG.

| Systematic name | Standard name | MAGIC phenotype |
| --- | --- | --- |
| YDR477W | SNF1 | Class 1 |
| YML016C | PPZ1 | Class 1 |
| YJR120W | | Class 1 |
| YOL055C | THI20 | Class 1 |
| YKL057C | NUP120 | Class 1 |
| **YML024W** | **RPS17A** | Class 2 |
| YDR083W | RRP8 | Class 2 |
| YCR002C | CDC10 | Class 2 |
| **YKL143W** | **LTV1** | Class 2 |
| YLL026W | HSP104 | Class 2 |
| YPR159W | KRE6 | Class 2 |
| **YOR096W** | **RPS7A** | Class 2 |
| YMR116C | ASC1 | Class 2 |
| YPR057W | BRR1 | Class 2 |
| YJR074W | MOG1 | Class 2 |
| YCR068W | ATG15 | Class 2 |
| YML062C | MFT1 | Class 2 |
| **YML026C** | **RPS18B** | Class 2 |
| YML013W | UBX2 | Class 2 |
| YMR032W | HOF1 | Class 2 |
| YNR029C | ZNG1 | Class 2 |
| YDL020C | RPN4 | Class 2 |
| YER151C | UBP3 | Class 2 |
| YMR255W | GFD1 | Class 2 |
| YMR307W | GAS1 | Class 2 |
| YOR035C | SHE4 | Class 2 |
| YOL072W | THP1 | Class 2 |
| **YDL083C** | **RPS16B** | Class 2 |
| YOR258W | YOR258W | Class 2 |
| YOL129W | VPS68 | Class 2 |
| YHR163W | SOL3 | Class 2 |
| YLR372W | ELO3 | Class 2 |
| YKL191W | DPH2 | Class 2 |
| YIR032C | DAL3 | Class 2 |
| YBR020W | GAL1 | Class 2 |
| **YJR145C** | **RPS4A** | Class 2 |
| YDR085C | AFR1 | Class 2 |
| YGR019W | UGA1 | Class 2 |

*Table 1 continued on next page*

*Table 1 continued*

| Systematic name | Standard name | MAGIC phenotype |
|---|---|---|
| YEL068C | | Class 2 |
| YIL112W | HOS4 | Class 2 |
| YKL198C | PTK1 | Class 2 |
| YER087C-A | | Class 2 |
| YJL200C | ACO2 | Class 2 |
| YJL160C | PIR5 | Class 2 |
| YMR034C | RCH1 | Class 2 |
| YGR132C | PHB1 | Class 2 |
| YLL033W | IRC19 | Class 2 |
| YGR072W | UPF3 | Class 2 |
| YGR016W | | Class 2 |
| YCR071C | IMG2 | Class 2 |
| YER060W | FCY21 | Class 2 |
| YER075C | PTP3 | Class 2 |
| YGR129W | SYF2 | Class 2 |
| YPR146C | | Class 2 |
| YEL012W | UBC8 | Class 2 |
| **YJR113C** | **RSM7** | Class 2 |
| **YPL173W** | **MRPL40** | Class 2 |
| YDL057W | | Class 2 |
| YBR068C | BAP2 | Class 2 |
| YHR200W | RPN10 | Class 2 |
| YOR298C-A | MBF1 | Class 2 |
| YER056C | FCY2 | Class 2 |
| **YNL081C** | **SWS2** | Class 2 |
| YGL114W | YGL114W | Class 2 |
| YAR030C | | Class 2 |
| YLR053C | NRS1 | Class 2 |
| YMR089C | YTA12 | Class 2 |
| YBR058C | UBP14 | Class 2 |
| YBR175W | SWD3 | Class 2 |
| YBR231C | SWC5 | Class 2 |
| YDR073W | SNF11 | Class 2 |
| **YDR115W** | **MRX14** | Class 2 |
| YGR136W | LSB1 | Class 2 |
| YGR159C | NSR1 | Class 2 |
| **YHL033C** | **RPL8A** | Class 2 |
| YHR011W | DIA4 | Class 2 |
| YHR143W | DSE2 | Class 2 |

*Table 1 continued on next page*

*Table 1 continued*

| Systematic name | Standard name | MAGIC phenotype |
| --- | --- | --- |
| YCL005W | LDB16 | Class 2 |
| YCL037C | SRO9 | Class 2 |
| YLR131C | ACE2 | Class 2 |
| YMR074C | SDD2 | Class 2 |
| YKL009W | MRT4 | Class 2 |
| YKL128C | PMU1 | Class 2 |
| YKL132C | RMA1 | Class 2 |
| YGR056W | RSC1 | Class 2 |
| YOR125C | CAT5 | Class 2 |
| YAL043C-a | | Class 2 |
| YLL015W | BPT1 | Class 2 |
| YOR235W | IRC13 | Class 2 |
| YJL179W | PFD1 | Class 2 |
| YLR387C | REH1 | Class 2 |
| **YLR388W** | **RPS29A** | Class 2 |
| YDR173C | ARG82 | Class 2 |
| YGL197W | MDS3 | Class 2 |
| YGL194C | HOS2 | Class 2 |
| YGL210W | YPT32 | Class 2 |
| YPL049C | DIG1 | Class 2 |
| YGL085W | LCL3 | Class 2 |
| YNL156C | NSG2 | Class 2 |
| YKL213C | DOA1 | Class 2 |
| YKR042W | UTH1 | Class 2 |
| **YKR057W** | **RPS21A** | Class 2 |
| YLR065C | SND2 | Class 2 |
| YIL043C | CBR1 | Class 2 |
| YIL049W | DFG10 | Class 2 |
| YIL088C | AVT7 | Class 2 |
| YIL054W | | Class 2 |
| YOL111C | MDY2 | Class 2 |
| YOL122C | SMF1 | Class 2 |
| YER091C | MET6 | Class 2 |
| YNL316C | PHA2 | Class 2 |
| YDL213C | NOP6 | Class 2 |
| YDR006C | SOK1 | Class 2 |
| **YDR025W** | **RPS11A** | Class 2 |
| YBR297W | MAL33 | Class 2 |
| YCR025C | | Class 2 |

*Table 1 continued on next page*

Table 1 continued

| Systematic name | Standard name | MAGIC phenotype |
|---|---|---|
| YML088W | UFO1 | Class 2 |
| YNL008C | ASI3 | Class 2 |
| YNL010W | PYP1 | Class 2 |
| YNR047W | FPK1 | Class 2 |
| YBR027C |  | Class 2 |
| YBR043C | QDR3 | Class 2 |
| YML036W | CGI121 | Class 2 |
| YPL004C | LSP1 | Class 2 |
| YML066C | SMA2 | Class 2 |
| YBR133C | HSL7 | Class 2 |
| YDL002C | NHP10 | Class 2 |
| YBR172C | SMY2 | Class 2 |
| YDL021W | GPM2 | Class 2 |
| **YDR462W** | **MRPL28** | Class 2 |
| **YDR500C** | **RPL37B** | Class 2 |
| YGL136C | MRM2 | Class 2 |
| YER174C | GRX4 | Class 2 |
| YER167W | BCK2 | Class 2 |
| YMR221C | FMP42 | Class 2 |
| YIL094C | LYS12 | Class 2 |
| YGR254W | ENO1 | Class 2 |
| YMR257C | PET111 | Class 2 |
| YMR278W | PRM15 | Class 2 |
| YMR291W | TDA1 | Class 2 |
| YMR303C | ADH2 | Class 2 |
| YNL303W |  | Class 2 |
| **YNL302C** | **RPS19B** | Class 2 |
| YNL265C | IST1 | Class 2 |
| YNL264C | PDR17 | Class 2 |

2% glucose) result in constitutive activation of Snf1 and relief from glucose repression of transcription (*Tu and Carlson, 1995*; *Ludin et al., 1998*; *Sanz et al., 2000*; *Ruiz et al., 2011*; *Caligaris et al., 2023*; *Figure 1I*). We found that Δ*reg1* cells exhibited significantly less accumulation of FlucSM in mitochondria than WT cells, and likewise, WT cells that grew in low glucose medium (LG: 0.1% glucose plus 3% glycerol) showed significantly lower FlucSM spGFP compared to cells in HG (*Figure 1G and H*). The absence of glycerol in LG (LG-Gly) did not cause any noticeable difference to LG (*Figure 1—figure supplement 2F and G*). Snf1 activation under these conditions was validated by the nuclear export of Mig1, which depends on phosphorylation by active Snf1 (*De Vit et al., 1997*; *Treitel et al., 1998*; *DeVit and Johnston, 1999*; *Figure 1—figure supplement 2H and I*). In addition, the abundance of FlucSM-GFP$_{11}$ induced by estradiol was not affected by Snf1 activation, and Grx5-GFP$_{1-10}$ level was unchanged in low glucose media and even elevated in Δ*reg1* cells – a trend opposite of the spGFP changes (*Figure 1—figure supplement 2J*). These data exclude the possibility that reduced expression

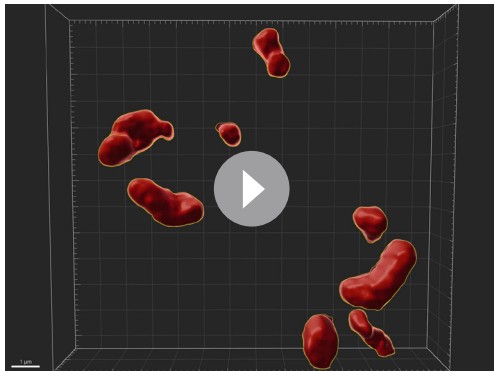

**Video 1.** 3D reconstructed structured illumination microscopy (SIM) images showing FlucSM split-GFP (spGFP) inside mitochondria after 90 min estradiol treatment. The mitochondrial outer membrane is labeled with Tom70-mCherry.

https://elifesciences.org/articles/87518/figures#video1

of either protein led to lower spGFP signal in mitochondria. To examine the effect of Snf1 activation on spGFP reconstitution, Grx5 spGFP strain was constructed in which the endogenous mitochondrial matrix protein Grx5 was C-terminally tagged with GFP$_{11}$ at its genomic locus, and GFP$_{1-10}$ was targeted to mitochondria through cleavable Su9 MTS (MTS-mCherry-GFP$_{1-10}$) (*Ruan et al., 2017*). Only modest reduction in Grx5 spGFP mean intensity was observed in LG compared to HG, and no significant difference after adjusting the GFP$_{1-10}$ abundance (spGFP/mCherry ratio) (*Figure 1—figure supplement 3A–D*). These data suggest that any effect on spGFP reconstitution is insufficient to explain the drastic reduction of MP accumulation in mitochondria under Snf1 activation. Overall, our results demonstrate that Snf1 activation primarily prevents mitochondrial accumulation of MPs, but not that of normal mitochondrial proteins.

We previously showed that the import of firefly luciferase mutants into mitochondria of human RPE-1 cells was positively correlated with protein instability (*Ruan et al., 2017*; *Gupta et al., 2011*). Using the established spGFP reporter, we found that treatment of RPE-1 cells with dorsomorphin, a chemical inhibitor of AMPK (*Zhou et al., 2001*), significantly increased mitochondrial accumulation of FlucDM (*Figure 1J and K*), but not GST, a well-folded protein control (*Figure 1L*). In contrast, pharmacological activation of AMPK via 5-aminoimidazole-4-carboxamide ribonucleoside (AICAR) (*Herrero-Martín et al., 2009*), significantly reduced FlucDM accumulation in mitochondria (*Figure 1J and M*). These results suggest that AMPK in human cells regulates MP accumulation in mitochondria following a similar trend as in yeast, although the underlying mechanisms might differ between these organisms.

## Mechanisms of MAGIC regulation by Snf1

The accumulation of MPs in mitochondria as observed using the spGFP reporter should depend on the relative rates of import versus degradation by mitochondrial proteases, most prominently Pim1 – the conserved Lon protease in yeast (*Ruan et al., 2017*). Three possible factors could therefore contribute to the reduced mitochondrial accumulation of MPs under Snf1 activation: (1) enhanced intra-mitochondrial degradation, (2) reduced cytosolic MP (due to enhanced folding and/or other degradation pathways), and (3) blocked mitochondrial import (*Figure 2A*). To evaluate the first possibility, an antimorphic mutant *pim1$^{S974D}$* was used to block the degradation of imported FlucSM in the mitochondrial matrix (*Nitika et al., 2022*). Indeed, in HG medium WT cells overexpressing *pim1$^{S974D}$* showed a significantly increased accumulation of FlucSM in mitochondria compared to cells overexpressing *PIM1* (*Figure 2B and C*). However, *pim1$^{S974D}$* overexpression was unable to increase FlucSM accumulation in mitochondria of *Δreg1* cells or WT cells growing in LG medium (*Figure 2B and C*). This result argued against the first possibility, and consistently the abundance of Pim1 protein was not increased by switching to nonfermentable carbon sources (*Morgenstern et al., 2017*). To evaluate the second possibility, we used an in vivo firefly luciferase assay (*Nathan et al., 1997*) and assessed the folding of enzymatically active FlucSM after estradiol induction. The result showed that Snf1-active cells exhibited reduced FlucSM luciferase activity, suggesting an increased rather than decreased fraction of misfolded FlucSM (*Figure 2D*). Furthermore, blocking the activated autophagy pathway in LG medium (*Iwama and Ohsumi, 2019*) did not increase FlucSM spGFP in mitochondria (*Figure 2—figure supplement 1A and B*). We also observed that proteasomal inhibition through MG132 treatment stimulated the mitochondrial accumulation of FlucSM but did not ablate the difference between HG and LG condition (*Figure 2—figure supplement 1C*). The stimulating effect of MG132 was not surprising because FlucSM is degraded by proteasome in the cytosol (*Ruan et al., 2017*) and

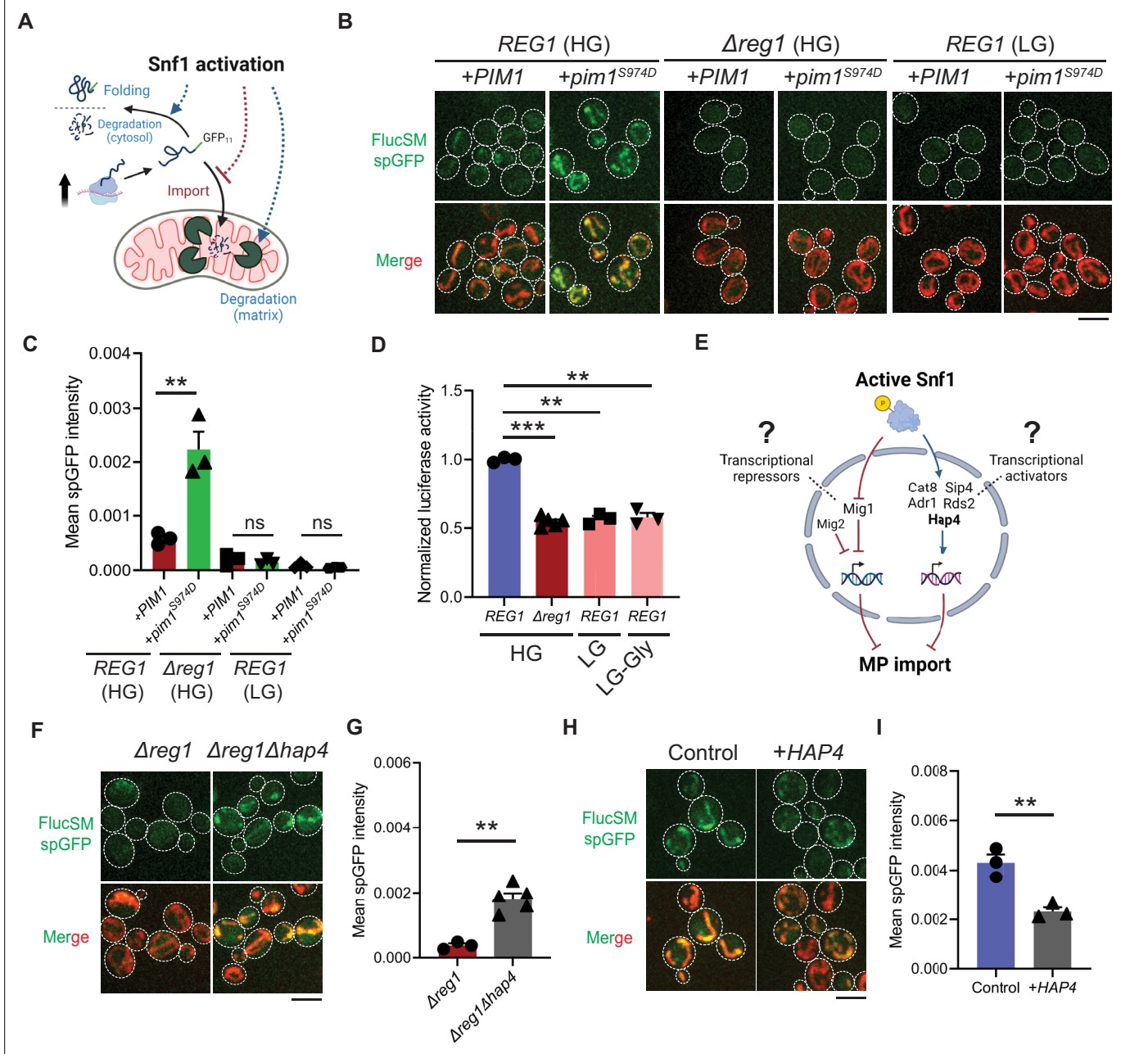

**Figure 2.** Snf1 negatively regulates mitochondrial import of cytosolic misfolded proteins (MPs). (**A**) Schematic diagram showing three possible explanations for reduced split-GFP (spGFP) in mitochondria of Snf1-active cells: reduced MPs, blocked import, or enhanced degradation. (**B, C**) Representative images (**B**) and quantification (**C**) of FlucSM spGFP in Snf1-inactive and Snf1-active cells overexpressing copper-inducible *PIM1* or *pim1^S974D^*. Shown in (**C**): means ± SEM of spGFP intensities (n=3). Unpaired two-tailed *t*-test. (**D**) Relative in vivo luciferase activity after 90 min of estradiol treatment. Means ± SEM of normalized FlucSM activity are shown (n=3 for *REG1*, n=5 for *Δreg1*). Paired (wild-type [WT] in different media) or unpaired (WT vs. *Δreg1* in HG) two-tailed *t*-test. LG-Gly: 0.1% glucose only. (**E**) Hypothetical regulations of import of MPs through transcriptional repressors and activators downstream of Snf1 activation. (**F, G**) Representative images (**F**) and quantification (**G**) of FlucSM spGFP in *Δreg1* and *Δreg1Δhap4* cells in HG medium. Shown in (**G**): means ± SEM of spGFP intensity (n=3 for *Δreg1*, n=5 for *Δreg1Δhap4*). Unpaired two-tailed *t*-test. (**H, I**) Representative images (**H**) and quantification (**I**) of FlucSM spGFP in WT cells (control) or with constitutive overexpression of *HAP4* in HG medium. Shown in (**I**): means ± SEM of spGFP intensities (n=3). Unpaired two-tailed *t*-test. **p<0.01; ***p<0.001; ns, not significant, p>0.05. Scale bars, 5 µm.

The online version of this article includes the following source data and figure supplement(s) for figure 2:

**Source data 1.** Split-GFP (spGFP) intensity and luciferase activity.

**Figure supplement 1.** Reduced accumulation of misfolded proteins in mitochondria under Snf1 activation is neither caused by elevated autophagy nor mediated by certain transcription factors.

**Figure supplement 1—source data 1.** Quantification of FlucSM split-GFP (spGFP) in different mutants or under drug treatment.

preventing this pathway could divert more of such protein molecules toward MAGIC. We thus favor the third possibility that Snf1 activation specifically prevents the import of MPs into mitochondria.

Next, we investigated downstream transcription factors that could mediate the Snf1-regulated MP import (*Figure 2E*). In the presence of abundant glucose and when Snf1 activity is low, transcriptional repressor Mig1 and its partially redundant homolog Mig2 are localized in the nucleus to confer glucose-repressed gene expression (*De Vit et al., 1997*; *Treitel et al., 1998*; *Westholm et al., 2008*). However, neither single deletion of *MIG1* nor double deletions of *MIG1* and *MIG2* reduced FlucSM spGFP in HG medium (*Figure 2—figure supplement 1D and E*), suggesting that Mig1 and/or Mig2-repressed gene expression was not sufficient to prevent MP import (*Figure 2E*, left branch). Then we tested if MP import was antagonized by transcriptional activators downstream of Snf1 including Cat8, Hap4, Sip4, Adr1, and Rds2 (*Hedbacker and Carlson, 2008*; *Gancedo, 1998*; *Schüller, 2003*; *Broach, 2012*; *Figure 2E*, right branch). Interestingly, only deletion of *HAP4*, but not other transcriptional activators, significantly rescued FlucSM import defect in *Δreg1* cells with Snf1 activation (*Figure 2F and G*; *Figure 2—figure supplement 1F and G*). When cultured in LG medium, *HAP4* deletion also resulted in a significant increase in mitochondrial accumulation of FlucSM in comparison to WT (*Figure 2—figure supplement 1H*). Furthermore, overexpression of Hap4 alone was sufficient to reduce FlucSM spGFP in HG medium (*Figure 2H and I*). These data suggest that Hap4 is a main downstream effector of Snf1 that regulates MP import.

Hap4 is the transcriptional activation subunit in the Hap2/3/4/5 complex that activates the expression of nuclear-encoded mitochondrial proteins and contributes to mitochondrial biogenesis during metabolic shifts or cellular aging (*Gancedo, 1998*; *Schüller, 2003*; *Broach, 2012*; *Forsburg and Guarente, 1989*; *Lin et al., 2002*). We hypothesized that elevated expression of mitochondrial preprotein induced by activation of Snf1-Hap4 axis (*Wright and Poyton, 1990*; *Morgenstern et al., 2017*; *Lin et al., 2002*; *von Plehwe et al., 2009*; *Hübscher et al., 2016*; *Di Bartolomeo et al., 2020*) may outcompete MPs for import channels, especially considering that previous studies have confirmed that the expression of TOM complex components on the mitochondrial outer membrane was static in Snf1-active cells (*Morgenstern et al., 2017*; *Di Bartolomeo et al., 2020*; *Figure 3A*).

To test this hypothesis, we attempted to restore MP import during Snf1 activation by using high-level expression of the soluble cytosolic domain of import receptors. The cytosolic import receptors lacking membrane-anchoring sequences are known to prevent mitochondrial preproteins from binding TOM complexes and thus inhibit preprotein import (*Brix et al., 1997*; *Brix et al., 2000*; *Schmidt et al., 2011*; *Figure 3—figure supplement 1A*). Interestingly, overexpression of the cytosolic domain of Tom70 (Tom70$_{cd}$), but not Tom20$_{cd}$ or Tom22$_{cd}$, significantly increased FlucSM import in LG medium (*Figure 3B and C*). Tom70$_{cd}$ also further increased FlucSM import in HG medium (*Figure 3—figure supplement 1B and C*). The effect of Tom70$_{cd}$ in cytosol required both the substrate binding and the chaperone-interaction domain (*Figure 3C*; *Figure 3—figure supplement 1D and E*). These results suggest that Tom70-dependent preprotein import may compete with MP import for limited TOM complexes. To further test if endogenous full-length Tom70 on the mitochondrial outer membrane is dispensable for MP import, we deleted *TOM70* and its paralog *TOM71* and found that in HG medium where mitochondrial respiration is not essential, FlucSM accumulation in mitochondria was not impaired in single mutants and increased in double mutant (*Figure 3D and E*). This result indicates that MP import does not use Tom70/Tom71 as obligatory receptors. The effect of *Δtom70Δtom71* on MP import was consistent, albeit less pronounced, with Tom70$_{cd}$ overexpression (*Figure 3D and E*; *Figure 3—figure supplement 1B and C*). One potential explanation for the modest effect in double mutant is that given to the functional redundancy between Tom20 and Tom70 (*Steger et al., 1990*; *Young et al., 2003*), Tom20 receptors in *Δtom70Δtom71* cells could instead mediate preprotein import, whereas cytosolic Tom70$_{cd}$ may have a dominant inhibitory effect on preprotein import by reducing association between preproteins and mitochondrial outer membrane or TOM complexes (*Brix et al., 1997*; *Brix et al., 2000*; *Schmidt et al., 2011*). Together, these data suggest that increased expression and receptor-dependent import of certain mitochondrial preproteins under Snf1 activation might indirectly restrict the import of MPs.

As the main entry gate for mitochondrial preproteins, the TOM complex adopts two functional conformations with different substrate specificity: the receptor-free dimer is primarily responsible for importing MIA pathway substrates and the receptor-bound trimer is for Tim23 pathway substrates (*Shiota et al., 2015*; *Araiso et al., 2019*; *Sakaue et al., 2019*). Deletion of Tom6 disassembles the

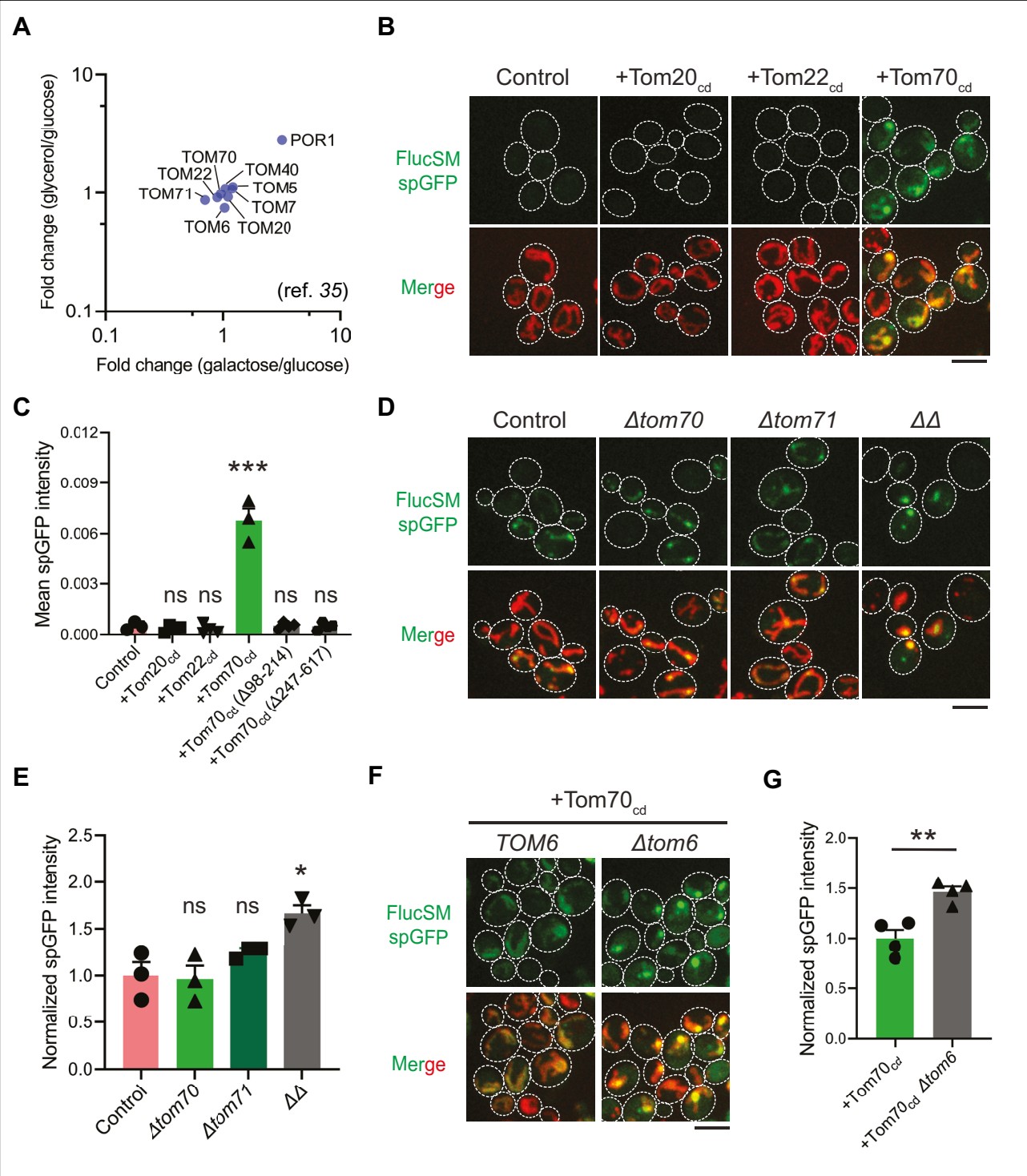

**Figure 3.** Mechanisms underlying Snf1-regulated misfolded protein (MP) import into mitochondria. (**A**) Fold changes in protein abundance of TOM complex components in glucose-limiting condition (glycerol or galactose) compared to glucose-rich condition. Raw data are retrieved from a published quantitative mass spectrometry dataset (***Morgenstern et al., 2017***). (**B, C**) Representative images (**B**) and quantification (**C**) of FlucSM split-GFP (spGFP) in wild-type control cells (n=3) and cells overexpressing $Tom20_{cd}$ (n=4), $Tom22_{cd}$ (n=4), and $Tom70_{cd}$ (n=3) (**C**), or truncated $Tom70_{cd}$ variants (n=4) (***Figure 3—figure supplement 1D***) in LG medium. Shown in (**C**): means ± SEM of spGFP intensities. Unpaired two-tailed *t*-test between control and overexpression strains. (**D, E**) Representative images (**D**) and quantification (**E**) of FlucSM spGFP in wild-type control, *Δtom70*, *Δtom71*, and *Δtom70 Δtom71* (*ΔΔ*) cells in HG medium. Shown in (**D**): top, FlucSM spGFP; bottom, merged images of spGFP and mitochondria labeled with mCherry-Fis1TM. Shown in (**E**): means ± SEM of normalized spGFP intensity (n=3). Unpaired two-tailed *t*-test. (**F, G**) Representative images (**F**) and quantification (**G**) of

*Figure 3 continued on next page*

*Figure 3 continued*

FlucSM spGFP in control and Δ*tom6* cells overexpressing Tom70$_{cd}$ in LG medium. Shown in (**G**): means ± SEM of normalized spGFP intensities (n=4). Unpaired two-tailed *t*-test. *p<0.05; **p<0.01; ***p<0.001; ns, not significant, p>0.05. Scale bars, 5 µm.

The online version of this article includes the following source data and figure supplement(s) for figure 3:

**Source data 1.** Previously reported mass spectrometry dataset and quantification of split-GFP (spGFP) in various mutants.

**Figure supplement 1.** Role of Tom70 cytosolic domain and Tom6 in regulating misfolded protein import.

**Figure supplement 1—source data 1.** Normalized split-GFP (spGFP) intensity.

**Figure supplement 1—source data 2.** Raw and labeled immunoblots for *Figure 3—figure supplement 1E*.

trimer and shifts the conformation equilibrium toward the dimer form (*Sakaue et al., 2019*; *Harbauer et al., 2014*). To test if the substrate selectivity of TOM complex regulates MP import, we eliminated the trimer conformation by deleting *TOM6* and found that it elevated FlucSM import in LG medium with or without Tom70$_{cd}$ overexpression (*Figure 3F and G*; *Figure 3—figure supplement 1F and G*). This result suggests that restricting MP import under Snf1 activation requires the trimeric TOM complex in addition to the competing mitochondrial preprotein import, and MPs might preferentially cross the mitochondrial outer membrane through the dimeric TOM complex.

## AMPK protects cellular fitness during proteotoxic stress

We next investigated the physiological effects of metabolic regulation of MAGIC mediated by Snf1/ AMPK. Prolonged induction of high-level FlucSM expression imposed a proteotoxic stress and led to a reduced growth rate in HG medium compared to the control, but interestingly no growth reduction was observed under glucose limitation (*Figure 4A*; *Figure 4—figure supplement 1A, D, and E*). We reasoned that the lack of growth defect in LG medium could be due to prevention of MP import into mitochondria downstream of Snf1 activation. Supporting this, elevating MP import by Tom70$_{cd}$ over-expression led to a reduced growth rate in LG medium that was dependent on FlucSM expression (*Figure 4A*; *Figure 4—figure supplement 1B*). Tom70$_{cd}$ overexpression also exacerbated growth rate reduction due to FlucSM expression in HG medium (*Figure 4A*; *Figure 4—figure supplement 1B*). In contrast, negative controls using truncated Tom70$_{cd}$ mutants that could not restore MP import did not produce the same growth defect (*Figure 4—figure supplement 1C*).

To further test whether the reduction in growth rate during proteotoxic stress was associated with impaired mitochondrial fitness, we assessed MMP using the dye tetramethylrhodamine methyl ester (TMRM). In HG medium and after 90 min induction of FlucSM, there was a negative relationship between spGFP accumulation and MMP: spGFP-positive cells exhibited a significantly reduced MMP level than spGFP-negative cells (*Figure 4C*). Again, this difference was not observed in cells that grew in LG, whereas Tom70$_{cd}$ overexpression led to a significant increase in the fraction of spGFP-positive cells with reduced MMP in both HG and LG medium (*Figure 4B and C*). These results suggest that Snf1 activation under glucose limitation protects mitochondrial and cellular fitness from FlucSM-associated proteotoxic stress.

Many neurodegenerative disease-associated aggregation-prone proteins, such as α-synuclein (*Devi et al., 2008*), FUS$^{P525L}$ (*Deng et al., 2018*; *Deng et al., 2015*), TDP-43 (*Wang et al., 2016*), amyloid beta (*Hansson Petersen et al., 2008*), and C9ORF72-associated poly(GR) dipeptide (*Choi et al., 2019*), are detected in mitochondria of human patients or disease models and impair mitochondrial functions. We wonder whether such toxic effects of disease-associated proteins can be counteracted by AMPK activation. First, we used the spGFP reporter in yeast and observed mitochondrial import of α-synuclein and FUS$^{P525L}$ in HG medium (*Figure 4D and E*; *Figure 4—figure supplement 1F and G*; *Video 2*). We found that Snf1 activation via glucose limitation or Δ*reg1* significantly reduced their accumulation in mitochondria, whereas Tom70$_{cd}$ overexpression reversed this effect (*Figure 4D–I*; *Figure 4—figure supplement 1H–K*). Mitochondrial import of α-synuclein and FUS$^{P525L}$ in HG medium was associated with lower MMP, and Tom70$_{cd}$ overexpression significantly increased the fraction of spGFP-positive and MMP-low cells in both HG and LG medium (*Figure 4J–K*). Furthermore, accumulation of α-synuclein in mitochondria correlated with a loss of respiratory capacity, as overexpression of Tom70$_{cd}$ and α-synuclein synergistically promoted the formation of respiration-deficient petite cells (*Figure 4L*).

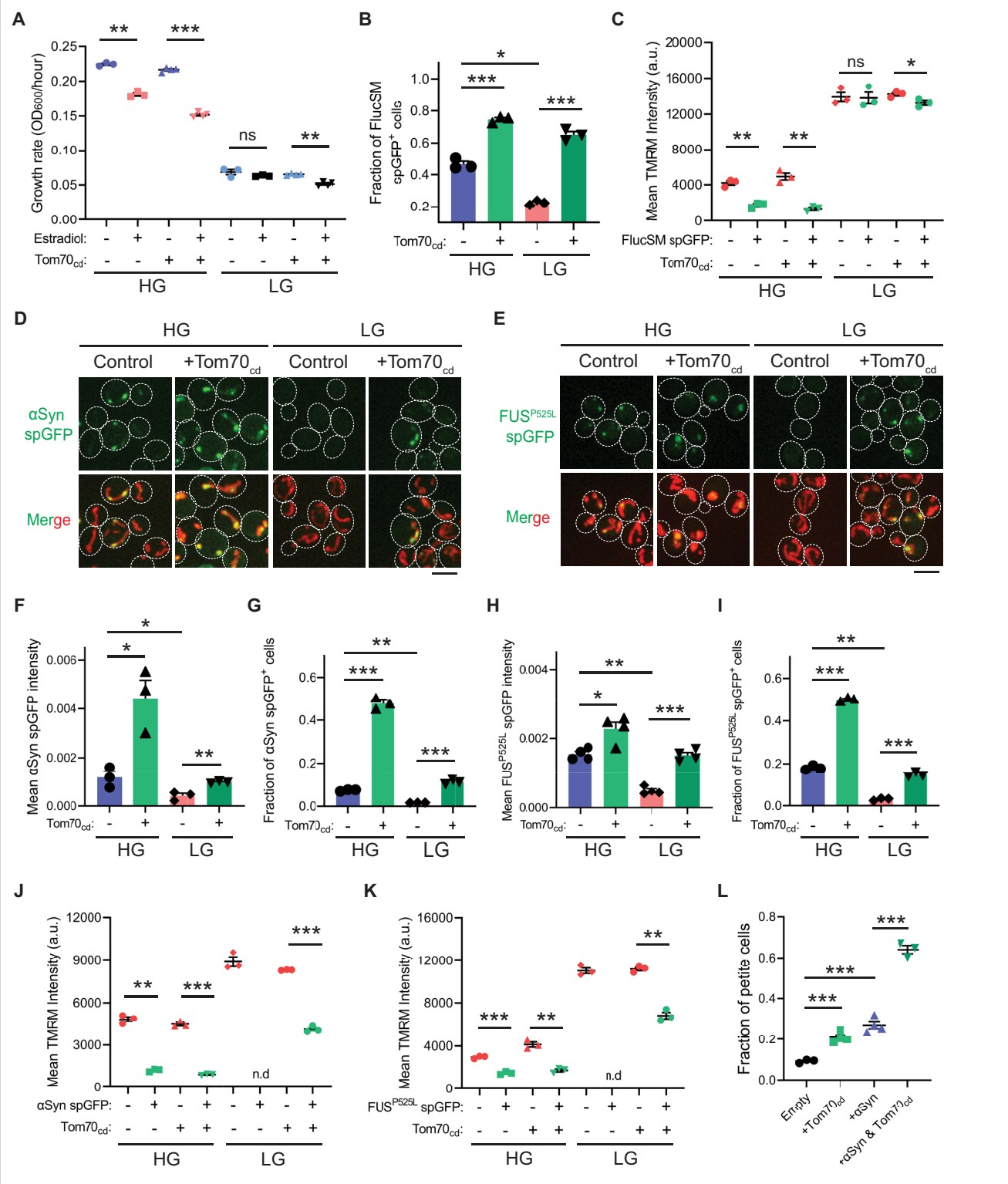

**Figure 4.** Snf1 activation protects cellular fitness against proteotoxic stress. (**A**) Growth rates of wild-type cells and cells overexpressing $Tom70_{cd}$ with (estradiol) or without (EtOH) FlucSM expression in HG and LG medium. Means ± SEM of $OD_{600}$ or growth rates are shown (n=3 for no $Tom70_{cd}$ expression, and n=4 for $Tom70_{cd}$ expression). Paired two-tailed *t*-test. (**B**) Fraction of FlucSM split-GFP (spGFP)-positive cells measured by flow cytometry. Means ± SEM are shown (n=3). Unpaired two-tailed *t*-test for cells growing in the same medium. Paired two-tailed *t*-test for control cells growing in different medium. (**C**) Comparisons of mitochondrial membrane potential between FlucSM spGFP-negative and spGFP-positive cells measured by tetramethylrhodamine methyl ester (TMRM). Means ± SEM are shown (n=3). Paired two-tailed *t*-test. (**D–I**) Representative images and

*Figure 4 continued on next page*

*Figure 4 continued*

quantifications of α-synuclein (αSyn) spGFP and FUS[P525L] spGFP signal. Shown in (**F, H**): means ± SEM of spGFP intensity measured by confocal imaging (n=3 for αSyn, and n=4 for FUS[P525L]). Shown in (**G, I**): means ± SEM of fraction of spGFP-positive cells measured by flow cytometry (n=3 for αSyn, and n=4 for FUS[P525L]). Unpaired two-tailed *t*-test for cells growing in the same medium. Paired two-tailed *t*-test for control cells between HG and LG medium. (**J, K**) Comparisons of membrane potential between αSyn or FUS[P525L] spGFP-negative and spGFP-positive cells measured by TMRM. Means ± SEM are shown (n=3 for αSyn, and n=4 for FUS[P525L]). Paired two-tailed *t*-test. n.d.: not determined due to limited positive cell counts in control cells growing in LG medium. (**L**) Fraction of respiratory-deficient petite cells measured by using tetrazolium overlay. Means ± SEM are shown (n=3 for empty control and αSyn with Tom70_cd, and n=4 for the rest). Unpaired two-tailed *t*-test. HG: 2% glucose; LG: 0.1% glucose plus 3% glycerol. *p<0.05; **p<0.01; ***p<0.001; ns, not significant, p>0.05. Scale bars, 5 µm.

The online version of this article includes the following source data and figure supplement(s) for figure 4:

**Source data 1.** Quantification of growth rate, split-GFP (spGFP), tetramethylrhodamine methyl ester (TMRM) intensity, and petite cell fraction.

**Figure supplement 1.** Snf1 activation protects against stress associated with FlucSM overexpression and prevents the accumulation of α-synuclein and FUS[P525L] in yeast mitochondria.

**Figure supplement 1—source data 1.** Raw data of growth curves, and quantification of growth rate and split-GFP (spGFP) intensity.

We next tested whether reducing mitochondrial accumulation of FUS[P525L] ameliorates its cellular toxicity in human cells. FUS[P525L] has been shown to bind mitochondrial Hsp60 and ATP synthase β-subunit to induce mitochondrial fragmentation and cell death (*Deng et al., 2018*; *Deng et al., 2015*). We expressed FUS[P525L] into human RPE-1 cells by transient transfection and confirmed the import of FUS[P525L] into mitochondrial matrix using the spGFP reporter (*Figure 5A and B*). FUS[P525] expression also caused the loss of MMP and elevated cell death compared to GST control (*Figure 5C and D*). Importantly, mitochondrial accumulation and fitness decline caused by FUS[P525] expression were significantly reduced by activation of AMPK via AICAR treatment (*Figure 5B–D*). These results suggest a protective role of AMPK in FUS-induced cellular toxicities possibly through preventing the import of the disease protein into mitochondria.

## Discussion

Metabolic imbalance and loss of proteostasis are interconnected hallmarks of aging and age-related diseases (*López-Otín et al., 2023*; *Hipp et al., 2019*; *Ottens et al., 2021*). Various metabolic signaling pathways, such as TOR, AMPK, Sirtuins, and insulin/IGF-1, sense metabolic stimuli, regulate cellular stress responses and influence major cytosolic protein quality control mechanisms including ubiquitin-proteasome pathway and autophagy (*Ottens et al., 2021*). Mitochondria, the central target of metabolic signaling and major hub of energy production, participate in proteostasis by importing of cytosolic MPs lacking canonical MTS via the MAGIC pathway (*Ruan et al., 2017*). Here, our unbiased genetic screen in yeast revealed an unexpected link between cellular metabolism and proteostasis through MAGIC. Our data established Snf1/AMPK as a key regulator of MP import, which balances the mitochondrial metabolic and proteostasis functions in response to glucose availability and protects mitochondrial fitness under proteotoxic stress (*Figure 5E*). We speculate that, when glucose level is high and cells rely on glycolysis for ATP production, mitochondria play a 'moonlighting role' in cellular proteostasis through MAGIC, a process dependent on mitochondrial import and proteostasis machineries including chaperones, mitochondrial translocons, and proteases (*Ruan et al., 2017*). On the other hand, when glucose is limited and cells rely on oxidative phosphorylation for ATP generation, Snf1/AMPK activation shuts down MAGIC and promotes import of essential mitochondrial preproteins, thus ensuring mitochondrial fitness and energy production.

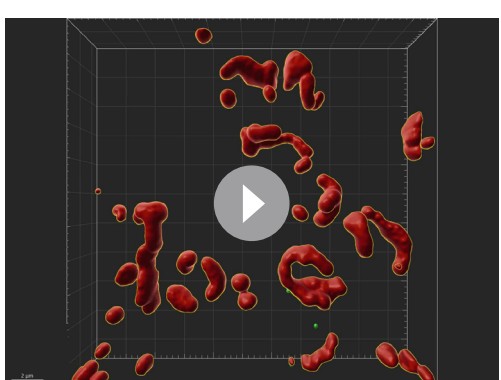

**Video 2.** 3D reconstructed structured illumination microscopy (SIM) images showing FUS[P525L] split-GFP (spGFP) inside mitochondria after 100 min estradiol treatment. The mitochondrial outer membrane is labeled with Tom70-mCherry.

https://elifesciences.org/articles/87518/figures#video2

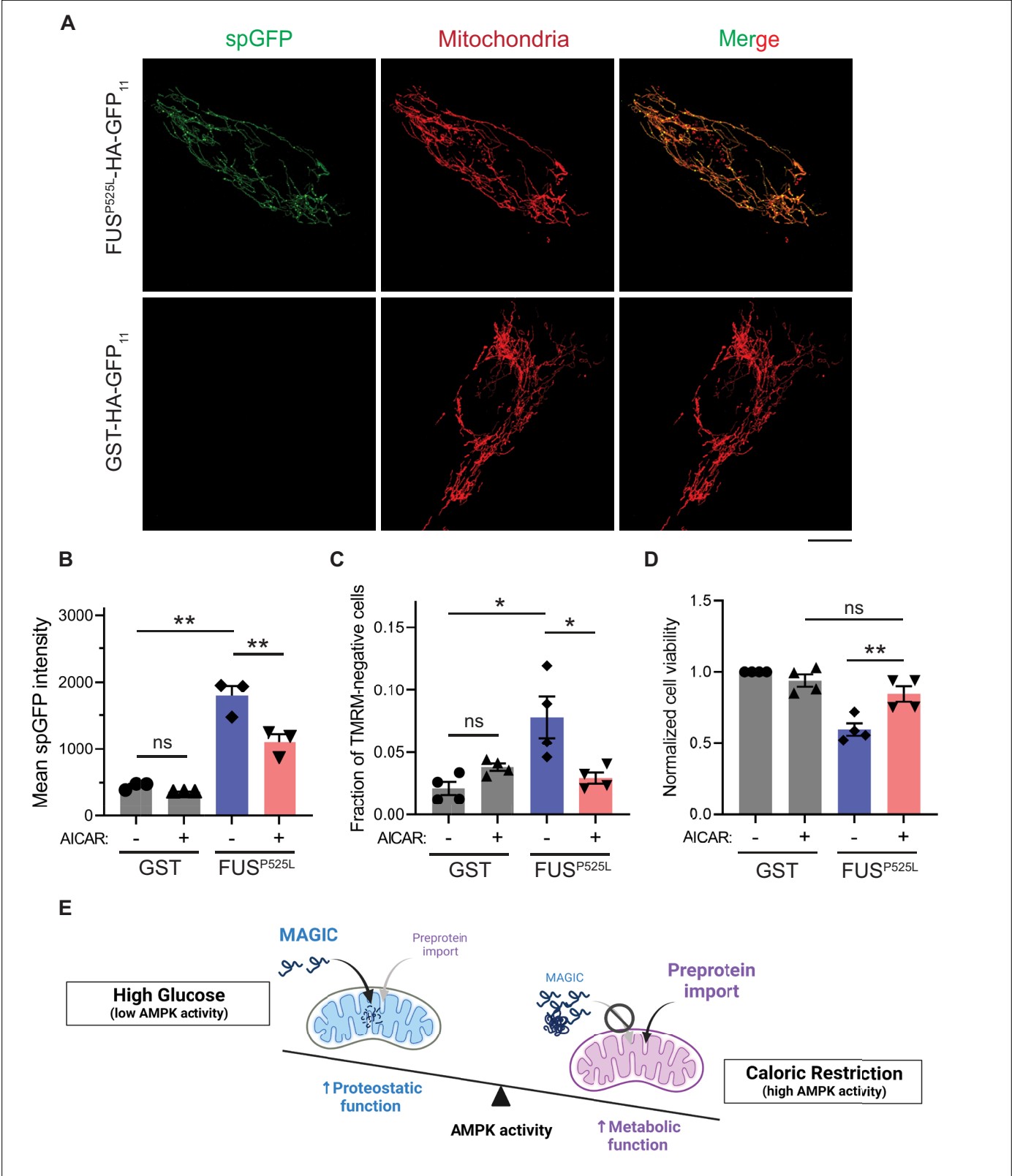

**Figure 5.** AMP-activated protein kinase (AMPK) activation prevents the accumulation of ALS-associated FUS[P525L] in mitochondria of RPE-1 cells and alleviates FUS-induced cytotoxicity. (**A, B**) Representative images (**A**) and flow cytometry quantification (**B**) of FUS[P525L] split-GFP (spGFP) and glutathione S-transferase (GST) spGFP in mitochondria of RPE-1 cells treated with or without 5-aminoimidazole-4-carboxamide ribonucleoside (AICAR). Shown in (**B**): means ± SEM of spGFP intensity (n=3). (**C, D**) Fraction of tetramethylrhodamine methyl ester (TMRM)-negative cells (**C**) and normalized cell viability

*Figure 5 continued on next page*

Figure 5 continued

(**D**) of RPE-1 cells expressing GST-HA-GFP$_{11}$ or FUS$^{P525L}$-HA-GFP$_{11}$ with or without AICAR treatment. Means ± SEM are shown (n=4). (**E**) Working model wherein Snf1/AMPK balances the metabolic and proteostasis function of mitochondria in response to glucose availability. Paired two-tailed *t*-test for the same cell line treated with drug or control medium. Unpaired two-tailed *t*-test between cell lines expressing GST and FUS$^{P525L}$. *p<0.05; **p<0.01; ns, not significant, p>0.05. Scale bars, 10 μm.

The online version of this article includes the following source data for figure 5:

**Source data 1.** Raw data for *Figure 5B–D*.

The downstream mechanism of this regulation remains to be fully elucidated. We propose that in yeast Snf1 activates the Hap4-dependent expression of mitochondrial preproteins which could compete with MPs for limited TOM complexes under glucose-limiting condition. Using cytosolic domain of Tom receptors to dampen preprotein import, we showed that only Tom70$_{cd}$ rescued MP import under Snf1 activation. A recent study (*Liu et al., 2022*) suggests that overexpression of full-length Tom70 leads to transcriptional activation for mitochondrial biogenesis. Whether the cytosolic Tom70$_{cd}$ fragment plays an indirect role in mitochondrial import through transcriptional regulation should be tested in the future. Since Snf1/Hap4 activation elevates the expression of hundreds of mitochondrial preproteins (*Morgenstern et al., 2017*; *Lin et al., 2002*; *von Plehwe et al., 2009*; *Hübscher et al., 2016*; *Di Bartolomeo et al., 2020*), it remains to be determined if specific preproteins or cytosolic factors are directly involved in inhibiting MP import. Furthermore, whether this metabolic control of MP import applies to other uncharacterized MAGIC substrates awaits further investigation.

Our data also suggest that the trimeric form of the TOM complex maintained by Tom6 is important for limiting MP entry under glucose restriction. We speculate that the receptor-binding state and substrate selectivity of different TOM conformations (*Sakaue et al., 2019*) could affect the permeability for MPs to enter mitochondria. Existing proteomic data suggest that the abundance of Tom6 is unaffected by Snf1 activation (*Morgenstern et al., 2017*; *Figure 3A*). As Tom6 can be phosphorylated by Cdk1 in a cell cycle-dependent manner (*Harbauer et al., 2014*), it may be interesting to investigate if Tom6 or other TOM complex components are targets of Snf1 kinase activity to directly modulate substrate specificity of the TOM complex.

A question raised by our findings is whether MAGIC is beneficial or detrimental to cells. Our data suggest that under physiological stress-free conditions, MP import and degradation in mitochondria is well tolerated, but an acute or chronic increase in the cytosolic MP load could overwhelm mitochondrial proteostasis capacity leading to organellar damage. If so, the regulation of MAGIC by AMPK could help explain the beneficial effect of caloric restriction on life span extension in model organisms (*Lin et al., 2002*; *Green et al., 2022*). In humans, the role of AMPK in health and diseases is complex and not fully understood (*Burkewitz et al., 2014*; *Steinberg and Kemp, 2009*; *Cantó et al., 2010*). While AMPK activity and mitochondrial gene expression mediated by downstream transcriptional factors such as PGC-1α and FOXO are elevated during health-benefitting activities such as exercise (*Cantó et al., 2010*), hyperactivated AMPK has also been reported in several neurodegenerative diseases with proteostasis decline (*Burkewitz et al., 2014*). Our findings suggest that elevating AMPK activity may be beneficial for alleviating proteotoxicity associated with degenerative diseases. Further studies using genetic approaches and relevant in vivo models could help elucidate the physiological role of AMPK in balancing proteostasis and mitochondrial fitness.

## Materials and methods
### Yeast strains, plasmids, and culture media
Yeast strains used in this study are based on the BY4741 strain background and listed in *Table 2*. Gene deletion and protein tagging were performed through PCR-mediated homologous recombination (*Longtine et al., 1998*) and verified by PCR genotyping. MAGIC YKO collection was constructed by incorporating MTS-mCherry-GFP$_{1-10}$ under GPD promoter into the TRP1 locus and tagging endogenous Lsg1 with GFP$_{11}$ in the YKO collection (*Giaever et al., 2002*). *Δreg1* and YKO strains harboring the deletion of the transcriptional factor downstream of Snf1 were freshly made and validated for at least three independent colonies.

**Table 2.** List of yeast strains and plasmids.

| Strain ID | Genotype | Source |
|---|---|---|
| BY4741 | *MATa his3Δ1; leu2Δ0; met15Δ0; ura3Δ0* | |
| RLY8616 | *GRX5-GFP₁₁-His3MX6; trp1::P_GPD-MTS-mCherry-GFP₁₋₁₀-natMX6* | **Ruan et al., 2017** |
| RLY8618 | *LSG1-GFP₁₁-His3MX6; trp1::P_GPD-MTS-mCherry-GFP₁₋₁₀-natMX6* | **Ruan et al., 2017** |
| RLY9798 | *LSG1-GFP₁₁-His3MX6; trp1::P_GPD-MTS-mCherry-GFP₁₋₁₀-natMX6; Δsnf1::kanMX6* | This study |
| RLY9799 | *LSG1-GFP₁₁-His3MX6; trp1::P_GPD-MTS-mCherry-GFP₁₋₁₀-natMX6; Δltv1::kanMX6* | This study |
| RLY9800 | *ura3Δ0::GEM-hphMX6; trp1::P_GPD-GRX5-HA-GFP₁₋₁₀-natMX6; HO::P_GAL1-FlucSM-HA-GFP₁₁-His3MX6; TOM70-mCherry-Ura3MX6* | This study |
| RLY9801 | *ura3Δ0::GEM-hphMX6; trp1::P_GPD-GRX5-HA-GFP₁₋₁₀-natMX6; HO::P_GAL1-FlucSM-HA-GFP₁₁-His3MX6; TOM70-mCherry-Ura3MX6; Δreg1::Leu2* | This study |
| RLY9802 | *trp1::P_GPD-GRX5-HA-GFP₁₋₁₀-natMX6; amp::GEM-P_GAL1-FlucSM-HA-GFP₁₁-kanMX6; TOM70-RFP-hphMX6* | This study |
| RLY9803 | *trp1::P_GPD-GRX5-HA-GFP₁₋₁₀-natMX6; amp::GEM-P_GAL1-FlucSM-HA-GFP₁₁-kanMX6; TOM70-mCherry-Ura3MX6* | This study |
| RLY9804 | *trp1::P_GPD-GRX5-HA-GFP₁₋₁₀-natMX6; amp::GEM-P_GAL1-FlucWT-HA-GFP₁₁-kanMX6; TOM70-mCherry-Ura3MX6* | This study |
| RLY9805 | *trp1::P_GPD-GRX5-HA-GFP₁₋₁₀-natMX6; amp::GEM-P_GAL1-FlucDM-HA-GFP₁₁-kanMX6; TOM70-mCherry-Ura3MX6* | This study |
| RLY9806 | *MIG1-GFP-His3MX6; PUS1-RFP-hphMX6* | This study |
| RLY9807 | *MIG1-GFP-His3MX6; PUS1-RFP-hphMX6; Δreg1::Leu2* | This study |
| RLY9808 | *ura3Δ0::P_CUP1-PIM1-Ura3; GRX5-GFP₁₋₁₀-natMX6; trp1::P_GPD-mCherry-Fis1TM-hphMX6; amp::GEM-P_GAL1-FlucSM-HA-GFP₁₁-kanMX6* | This study |
| RLY9809 | *ura3Δ0::P_CUP1-pim1^{S974D}-Ura3; GRX5-GFP₁₋₁₀-natMX6; trp1::P_GPD-mCherry-Fis1TM-hphMX6; amp::GEM-P_GAL1-FlucSM-HA-GFP₁₁-kanMX6* | This study |
| RLY9810 | *trp1::P_GPD-MTS-mCherry-natMX6; amp::GEM-P_GAL1-FlucSM-HA-GFP₁₁-kanMX6* | This study |
| RLY9811 | *trp1::P_GPD-MTS-mCherry-natMX6; amp::GEM-P_GAL1-FlucSM-HA-GFP₁₁-kanMX6; Δreg1::His3MX6* | This study |
| RLY9812 | *ura3Δ0::GEM-hphMX6; trp1::P_GPD-GRX5-HA-GFP₁₋₁₀-natMX6; HO::P_GAL1-FlucSM-HA-GFP₁₁-His3MX6; TOM70-mCherry-Ura3MX6; Δreg1::Leu2; Δhap4::kanMX6* | This study |
| RLY9813 | *trp1::P_GPD-GRX5-HA-GFP₁₋₁₀-natMX6; amp::GEM-P_GAL1-FlucSM-HA-GFP₁₁-kanMX6; TOM70-mCherry-Ura3MX6; HO::P_GPD-HAP4-hphMX6* | This study |
| RLY9814 | *ura3Δ0::GEM-hphMX6; trp1::P_GPD-GRX5-HA-GFP₁₋₁₀-natMX6; HO::P_GAL1-FlucSM-HA-GFP₁₁-His3MX6; TOM70-mCherry-Ura3MX6; Δatg1::kanMX6* | This study |
| RLY9815 | *ura3Δ0::GEM-hphMX6; trp1::P_GPD-GRX5-HA-GFP₁₋₁₀-natMX6; HO::P_GAL1-FlucSM-HA-GFP₁₁-His3MX6; TOM70-mCherry-Ura3MX6; Δatg15::kanMX6* | This study |
| RLY9816 | *ura3Δ0::GEM-hphMX6; trp1::P_GPD-GRX5-HA-GFP₁₋₁₀-natMX6; HO::P_GAL1-FlucSM-HA-GFP₁₁-His3MX6; TOM70-mCherry-Ura3MX6; Δmig1::kanMX6* | This study |
| RLY9817 | *ura3Δ0::GEM-hphMX6; trp1::P_GPD-GRX5-HA-GFP₁₋₁₀-natMX6; HO::P_GAL1-FlucSM-HA-GFP₁₁-His3MX6; TOM70-mCherry-Ura3MX6; Δmig2::Leu2* | This study |
| RLY9818 | *ura3Δ0::GEM-hphMX6; trp1::P_GPD-GRX5-HA-GFP₁₋₁₀-natMX6; HO::P_GAL1-FlucSM-HA-GFP₁₁-His3MX6; TOM70-mCherry-Ura3MX6; Δmig1::kanMX6; Δmig2::Leu2* | This study |
| RLY9819 | *ura3Δ0::GEM-hphMX6; trp1::P_GPD-GRX5-HA-GFP1-10-natMX6; HO::PGAL1-FlucSM-HA-GFP11-His3MX6; TOM70-mCherry-Ura3MX6; Δreg1::Leu2; Δcat8::kanMX6* | This study |
| RLY9820 | *ura3Δ0::GEM-hphMX6; trp1::P_GPD-GRX5-HA-GFP1-10-natMX6; HO::PGAL1-FlucSM-HA-GFP11-His3MX6; TOM70-mCherry-Ura3MX6; Δreg1::Leu2; Δsip4::kanMX6* | This study |
| RLY9821 | *ura3Δ0::GEM-hphMX6; trp1::P_GPD-GRX5-HA-GFP1-10-natMX6; HO::PGAL1-FlucSM-HA-GFP11-His3MX6; TOM70-mCherry-Ura3MX6; Δreg1::Leu2; Δrds2::kanMX6* | This study |
| RLY9822 | *ura3Δ0::GEM-hphMX6; trp1::P_GPD-GRX5-HA-GFP1-10-natMX6; HO::PGAL1-FlucSM-HA-GFP11-His3MX6; TOM70-mCherry-Ura3MX6; Δreg1::Leu2; Δadr1::kanMX6* | This study |

*Table 2 continued on next page*

*Table 2 continued*

| Strain ID | Genotype | Source |
|---|---|---|
| RLY9823 | trp1::P_GPD-GRX5-HA-GFP_{1-10}-natMX6; amp::GEM-P_GAL1-FlucSM-HA-GFP_{11}-kanMX6; TOM70-mCherry-Ura3MX6; HO::P_GPD-tom20_cd-hphMX6 | This study |
| RLY9824 | trp1::P_GPD-GRX5-HA-GFP_{1-10}-natMX6; amp::GEM-P_GAL1-FlucSM-HA-GFP_{11}-kanMX6; TOM70-mCherry-Ura3MX6; HO::P_GPD-tom22_cd-hphMX6 | This study |
| RLY9825 | ura3Δ0::GEM-hphMX6; trp1::P_GPD-GRX5-HA-GFP_{1-10}-natMX6; HO::P_GAL1-FlucSM-HA-GFP_{11}-His3MX6; TOM70-mCherry-Ura3MX6; amp::P_GPD-tom70_cd-3xFLAG-kanMX6 | This study |
| RLY9826 | ura3Δ0::GEM-hphMX6; trp1::P_GPD-GRX5-HA-GFP_{1-10}-natMX6; HO::P_GAL1-FlucSM-HA-GFP_{11}-His3MX6; TOM70-mCherry-Ura3MX6; amp::P_GPD-tom70_cd(Δ98–214)–3xFLAG-kanMX6 | This study |
| RLY9827 | ura3Δ0::GEM-hphMX6; trp1::P_GPD-GRX5-HA-GFP_{1-10}-natMX6; HO::P_GAL1-FlucSM-HA-GFP_{11}-His3MX6; TOM70-mCherry-Ura3MX6; amp::P_GPD-tom70_cd(Δ247–617)–3xFLAG-kanMX6 | This study |
| RLY9828 | ura3Δ0::GEM-hphMX6; trp1::P_GPD-GRX5-HA-GFP_{1-10}-natMX6; HO::P_GAL1-FlucSM-HA-GFP_{11}-His3MX6; trp1::P_GPD-mCherry-Fis1TM-kanMX6 | This study |
| RLY9829 | ura3Δ0::GEM-hphMX6; trp1::P_GPD-GRX5-HA-GFP_{1-10}-natMX6; HO::P_GAL1-FlucSM-HA-GFP_{11}-His3MX6; trp1::P_GPD-mCherry-Fis1TM-kanMX6; Δtom70::Ura3MX6 | This study |
| RLY9830 | ura3Δ0::GEM-hphMX6; trp1::P_GPD-GRX5-HA-GFP_{1-10}-natMX6; HO::P_GAL1-FlucSM-HA-GFP_{11}-His3MX6; trp1::P_GPD-mCherry-Fis1TM-kanMX6; Δtom71::Leu2 | This study |
| RLY9831 | ura3Δ0::GEM-hphMX6; trp1::P_GPD-GRX5-HA-GFP_{1-10}-natMX6; HO::P_GAL1-FlucSM-HA-GFP_{11}-His3MX6; trp1::P_GPD-mCherry-Fis1TM-kanMX6; Δtom70::Ura3MX6; Δtom71::Leu2 | This study |
| RLY9832 | ura3Δ0::GEM-hphMX6; trp1::P_GPD-GRX5-HA-GFP_{1-10}-natMX6; HO::P_GAL1-FlucSM-HA-GFP_{11}-His3MX6; TOM70-mCherry-Ura3MX6; amp::P_GPD-tom70_cd-3xFLAG-kanMX6; Δtom6::Leu2 | This study |
| RLY9833 | ura3Δ0::GEM-hphMX6; trp1::P_GPD-GRX5-HA-GFP_{1-10}-natMX6; HO::P_GAL1-FlucSM-HA-GFP_{11}-His3MX6 | This study |
| RLY9834 | ura3Δ0::GEM-hphMX6; trp1::P_GPD-GRX5-HA-GFP_{1-10}-natMX6; HO::P_GAL1-FlucSM-HA-GFP_{11}-His3MX6; amp::P_GPD-tom70_cd-3xFLAG-kanMX6 | This study |
| RLY9835 | ura3Δ0::P_GPD-a-Synuclein-HA-GFP_{11}-His3MX6; GRX5-GFP_{1-10}-natMX6; TOM70-mCherry-Ura3MX6 | This study |
| RLY9836 | ura3Δ0::P_GPD-a-Synuclein-HA-GFP_{11}-His3MX6; GRX5-GFP_{1-10}-natMX6; TOM70-mCherry-Ura3MX6; Δreg1::Leu2 | This study |
| RLY9837 | ura3Δ0::P_GPD-a-Synuclein-HA-GFP11-His3MX6; GRX5-GFP1-10-natMX6; TOM70-mCherry-Ura3MX6; trp1::P_GPD-tom70_cd-3xFLAG-kanMX6 | This study |
| RLY9838 | ura3Δ0::P_GPD-a-Synuclein-HA-GFP_{11}-His3MX6; GRX5-GFP_{1-10}-natMX6 | This study |
| RLY9839 | ura3Δ0::P_GPD-a-Synuclein-HA-GFP11-His3MX6; GRX5-GFP1-10-natMX6; trp1::P_GPD-tom70_cd-3xFLAG-kanMX6 | This study |
| RLY9840 | trp1::P_GPD-GRX5-HA-GFP_{1-10}-natMX6; TOM70-mCherry-Ura3MX6; amp::GEM-P_GAL1-FUS^{P525L}-HA-GFP_{11}-kanMX6 | This study |
| RLY9841 | ura3Δ0::GEM-hphMX6; trp1::P_GPD-GRX5-HA-GFP_{1-10}-natMX6; HO::P_GAL1-FUS^{P525L}-HA-GFP_{11}-His3MX6; TOM70-mCherry-Ura3MX6 | This study |
| RLY9842 | ura3Δ0::GEM-hphMX6; trp1::P_GPD-GRX5-HA-GFP_{1-10}-natMX6; HO::P_GAL1-FUS^{P525L}-HA-GFP_{11}-His3MX6; TOM70-mCherry-Ura3MX6; Δreg1::Leu2 | This study |
| RLY9843 | ura3Δ0::GEM-hphMX6; trp1::P_GPD-GRX5-HA-GFP_{1-10}-natMX6; HO::P_GAL1-FUS^{P525L}-HA-GFP_{11}-His3MX6; TOM70-mCherry-Ura3MX6; amp::P_GPD-tom70_cd-3xFLAG-kanMX6 | This study |
| RLY9844 | trp1::P_GPD-GRX5-HA-GFP_{1-10}-natMX6; amp::GEM-P_GAL1-FUS^{P525L}-HA-GFP_{11}-kanMX6 | This study |
| RLY9845 | trp1::P_GPD-GRX5-HA-GFP_{1-10}-natMX6; amp::GEM-P_GAL1-FUS^{P525L}-HA-GFP_{11}-kanMX6; trp1::P_GPD-tom70_cd-3xFLAG-kanMX6 | This study |
| RLY9846 | ura3Δ0::GEM-hphMX6; trp1::P_GPD-GRX5-HA-GFP_{1-10}-natMX6; HO::P_GAL1-FlucSM-HA-GFP_{11}-His3MX6; TOM70-mCherry-Ura3MX6; Δpdr5::kanMX6 | This study |
| RLY9847 | ura3Δ0::GEM-hphMX6; trp1::P_GPD-GRX5-HA-GFP_{1-10}-natMX6; HO::P_GAL1-FlucSM-HA-GFP_{11}-His3MX6; TOM70-mCherry-Ura3MX6; Δhap4::kanMX6 | This study |

| Plasmid ID | Construct | | Vector type | Source |
|---|---|---|---|---|

*Table 2 continued on next page*

*Table 2 continued*

| Strain ID | Genotype | Source | |
|---|---|---|---|
| RLB918 | $TRP1::P_{GPD}$-MTS-mCherry-$GFP_{1-10}$-natMX6 | Yeast expression | *Ruan et al., 2017* |
| RLB919 | $TRP1::P_{GPD}$-Grx5-HA-$GFP_{1-10}$-natMX6 | Yeast expression | *Ruan et al., 2017* |
| pJW1663 | GEM-$P_{GAL1}$-GFP-kanMX6 | Yeast expression | *Costa et al., 2018* |
| RLB1050 | $TRP1::P_{GPD}$-mCherry-Fis1TM-KanMX6 | Yeast expression | *Ruan et al., 2017* |
| RLB1051 | GEM-$P_{GAL1}$-FlucSM-HA-$GFP_{11}$-KanMX6 | Yeast expression | This study |
| RLB1052 | GEM-$P_{GAL1}$-FlucWT-HA-$GFP_{11}$-KanMX6 | Yeast expression | This study |
| RLB1053 | GEM-$P_{GAL1}$-FlucDM-HA-$GFP_{11}$-KanMX6 | Yeast expression | This study |
| RLB1054 | GEM-$P_{GAL1}$-$FUS^{P525L}$-HA-$GFP_{11}$-KanMX6 | Yeast expression | This study |
| RLB1055 | pRS316-$P_{CUP1}$-PIM1-Ura3 | Yeast expression | *Nitika et al., 2022* |
| RLB1056 | pRS316-$P_{CUP1}$-$pim1^{S974D}$-Ura3 | Yeast expression | *Nitika et al., 2022* |
| RLB1057 | pRS313-HO(homology)-$P_{GAL1}$-FlucSM-HA-$GFP_{11}$-His3MX6-HO(homology) | Yeast expression | This study |
| RLB1058 | pRS313-HO(homology)-$P_{GAL1}$-$FUS^{P525L}$-HA-$GFP_{11}$-His3MX6-HO(homology) | Yeast expression | This study |
| RLB1059 | pRS316-HO(homology)-$P_{GPD}$-HAP4-hphMX6-HO(homology) | Yeast expression | This study |
| RLB1060 | $TRP1::P_{GPD}$-$tom70_{cd}$-3xFLAG-KanMX6 | Yeast expression | This study |
| RLB1061 | $TRP1::P_{GPD}$-$tom70_{cd}$($\Delta$98–214)–3xFLAG-KanMX6 | Yeast expression | This study |
| RLB1062 | $TRP1::P_{GPD}$-$tom70_{cd}$($\Delta$247–617)–3xFLAG-KanMX6 | Yeast expression | This study |
| RLB1063 | pRS316-HO(homology)-$P_{GPD}$-$tom20_{cd}$-hphMX6-HO(homology) | Yeast expression | This study |
| RLB1064 | pRS316-HO(homology)-$P_{GPD}$-$tom22_{cd}$-hphMX6-HO(homology) | Yeast expression | This study |
| RLB1065 | $P_{GPD}$-a-Synuclein-HA-$GFP_{11}$-His3MX6 | Yeast expression | This study |
| RLB1066 | $P_{CMV}$-$FUS^{P525L}$-HA-$GFP_{11}$ | Mammalian expression | *Ruan et al., 2017* |
| RLB912 | $P_{CMV}$-MTS-mCherry-$GFP_{1-10}$ | Mammalian expression | *Ruan et al., 2017* |
| RLB914 | $P_{CMV}$-FlucDM-HA-$GFP_{11}$ | Mammalian expression | *Ruan et al., 2017* |
| RLB916 | $P_{CMV}$-GST-HA-$GFP_{11}$ | Mammalian expression | *Ruan et al., 2017* |

Human α-synuclein tagged with GFP11 under GPD promoter was cloned and inserted into the *ura3Δ0* locus. FlucSM-HA-$GFP_{11}$ and $FUS^{P525L}$-HA-$GFP_{11}$ under GAL1 promoter were cloned from plasmids from our previous study (*Ruan et al., 2017*) and plasmid 416Gal-FUS-P525L-YFP (Addgene plasmid #29628). FlucWT-HA-$GFP_{11}$ and FlucDM-HA-$GFP_{11}$ plasmids were constructed using site-directed mutagenesis kit (NEB) based on FlucSM-HA-$GFP_{11}$. Both $GFP_{11}$-tagged Fluc proteins and GEM transcriptional factor (cloned from pJW1663, Addgene plasmid #112037) were stably integrated into yeast genome. $GFP_{1-10}$ was fused with the mitochondrial matrix protein Grx5 under GPD promoter, except in experiments involving *PIM1* or $pim1^{S974D}$ mutant and α-synuclein spGFP where $GFP_{1-10}$ was fused to the C-terminus of endogenous Grx5 to avoid signal saturation. WT *PIM1* or $pim1^{S974D}$ mutant under CUP1 promoter, *HAP4*, cytosolic domain of Tom20 (1–97 aa), Tom22 (38–617 aa), Tom70 (38–617 aa), and truncated variants of Tom70cd under GPD promoter were cloned and stably integrated into yeast genome. Mitochondrial outer membrane was labeled with Tom70-mCherry

or Tom70-RFP, except for the Tom70/71 deletion experiments in which mitochondria were labeled with mCherry-Fis1TM (*Zhou et al., 2014*).

MAGIC YKO library construction, flow cytometry, and imaging during high-throughput screen were performed with synthetic defined minus histidine (SD-His) medium. Synthetic complete (SC) supplemented with 2% glucose (HG), 0.1% glucose plus 3% glycerol (LG), or 0.1% glucose (LG-Gly) was used for confocal imaging, luciferase assays, biochemistry, and TMRM staining. YEP medium (yeast extract-peptone) supplemented with 2% glucose (HG) or 0.1% glucose plus 3% glycerol (LG) was used for growth assays. Optical density at 600 nm ($OD_{600}$) was used to estimate the amount of yeast cells used in the various experiments.

## Drug treatments

β-Estradiol (E2758, MilliporeSigma, Burlington, MA, USA) was dissolved in $H_2O$ and added at a final concentration of 1 µM. $CuSO_4$ (C1297, MilliporeSigma) was dissolved in $H_2O$ and added at a final concentration of 0.5 mM. D-luciferin potassium salt (LUCK, GoldBio, St Louis, MO, USA) was freshly dissolved in appropriate yeast media at a final concentration of 0.5 mM. Dorsomorphin (S7840, Selleck Chemicals, Houston, TX, USA; 11967, Cayman Chemical, Ann Arbor, MI, USA) dissolved in DMSO was added to RPE-1 cells at the final concentration of 10 µM for 24 hr (*Li et al., 2019a*). AICAR was dissolved in DMSO (S1802, Selleck Chemicals) and added at the final concentration of 2 mM for 48 hr in the FlucDM experiment, or dissolved directly in media at the concentration of 2 mM (10010241, Cayman Chemical) for the $FUS^{P525L}$ experiment (*Robert et al., 2009*). MG132 (C2211, MilliporeSigma) was dissolved in DMSO and added to YEP-based medium at a final concentration of 80 µM.

## Yeast library construction and genome-wide screen

MAGIC YKO was constructed with a two-step transformation using the Frozen-EZ Yeast Transformation II Kit (T2001, Zymo Research, Irvine, CA, USA) following the microscale protocol in 96-well format. First, knockout strains were grown to saturation in deep-well plates containing 1 ml of YPD broth with G418 (200 µg/ml, Corning Inc, Corning, NY, USA). 150 µl of refreshed mid-log phase cultures and 0.2 µg of MTS-mCherry-GFP$_{1-10}$-clonNat DNA were used in the transformation setup on the epMotion 5075 liquid handling workstation (Eppendorf, Hamburg, Germany). To optimize transformation efficiency, the transformation mixtures were incubated for 2 hr and at the end of transformation they were transferred into deep-well plates with 4 volumes of YPD for 2 hr of outgrowth at 30°C. The transformants were selected for 4–5 days in 1 ml of YPD broth with clonNAT (200 µg/ml, GoldBio), resulting in the intermediate MTS-mCherry-GFP$_{1-10}$-clonNat library. Then the Lsg1-HA-GFP$_{11}$ tagging PCR product was integrated into the genome of the intermediate strains following the same protocol, with the exception that the finial library was selected in SD-His medium.

Total 4645 YKO strains with Lsg1 spGFP reporter were cultured in 96-well plates, and spGFP intensities before and after heat shock (30 min at 42°C) were measured at 488 nm excitation with appropriate filters on Attune NxT flow cytometer equipped with an auto sampler (Thermo Fisher Scientific, Waltham, MA, USA). After subtracting background from the populational mean spGFP intensity, KOs displaying different spGFP pattern were determined by a cutoff (smaller than 1.1-fold increase after heat shock) and further validated by live-cell confocal imaging. Based on the phenotype of mitochondrial spGFP intensity of each mutant at two imaging time points, Class 1 mutants were determined by the p value of comparing the spGFP/mCherry ratio of each single cell between KO and WT at permissive temperature, $p<0.01$, and Class 2 mutants were determined by the p value of comparing the spGFP intensity of each single cell of before and after heat shock for the same mutant, $p>0.01$. Genes involved in known mitochondrial import pathways were excluded from analysis.

## Confocal microscopy and imaging conditions

Live-cell images were acquired using a Yokogawa CSU-10 spinning disc on the side port of a Carl Zeiss 200 m inverted microscope or a Carl Zeiss LSM-780 confocal system. Laser 488 or 561 nm excitation was applied to excite GFP or mCherry, respectively, and the emission was collected through the appropriate filters onto a Hamamatsu C9100-13 EMCCD on the spinning disc confocal system or the single-photon avalanche photodiodes on the Zeiss 780 system. Regarding the multi-track acquisition, the configuration of alternating excitation was used to avoid the bleed-through of GFP (for dual-color imaging, GFP or mCherry labeled controls were applied for laser and exposure settings). The spinning

disc and the LSM780 were equipped with a 100×1.45 NA Plan-Apochromat objective and a 63×1.4 oil Plan-Apochromat objective, respectively. For yeast 3D imaging, 0.5 µm step size for 6 µm in total in Z; for human cells, 1 µm step size. Images were acquired using MetaMorph (version 7.0, MDS Analytical Technologies/Danaher, Sunnyvale, CA, USA) on the CSU-10 spinning disc system and Carl Zeiss ZEN software on the LSM780.

Yeast culture condition for imaging: yeast cells were cultured in SC or SD-His with appropriate carbon source overnight at 30°C. The cells were then refreshed in the corresponding medium for at least 3 hr at 30°C until reaching an $OD_{600}$ of about 0.2. For estradiol-GEM inducible systems, 1 µM of β-estradiol was added to the medium for 90 min unless indicated otherwise. For copper-inducible overexpression of *PIM1* or its mutant, 0.5 mM $CuSO_4$ was added for 2 hr, followed by the estradiol induction for 2 hr. All images in the same experiments were acquired with the same laser and exposure settings. Image processing was performed using ImageJ software (NIH, Bethesda, MD, USA) or Imaris software (Oxford Instruments Group, Abingdon, UK). For visualization purposes, images were scaled with bilinear interpolation and shown as the maximum projection on Z for fluorescent channels. Cell boundaries were delineated according to white-field images.

## SpGFP quantification

SpGFP fluorescence from confocal images was quantified by using a custom Python code described previously (*Ruan et al., 2017*), which can be found within the GitHub repository at https://github.com/RongLiLab/Wang-et-al.-2022.git (*Wang, 2022*). In brief, mCherry and GFP intensities were summed along the z-axis, and then subjected to a random walk segmentation of the background and water-shed segmentation of adjoining cells. For each cell, the mCherry channel was thresholded at 5% of maximal value to detect mitochondria, and median GFP intensity within mitochondria was calculated as spGFP intensity per cell. In the YKO imaging validation, Lsg1 spGFP/mCherry ratio of each cell was used for statistical analyses. For Lsg1 spGFP signal detected in *Δsnf1*, *Δltv1*, and *WT* cells, populational means spGFP/mCherry of at least three biological repeats were calculated. Adjusting Lsg1 spGFP intensity to mitochondrial mCherry intensity avoided the potential effect of changing local abundance of $GFP_{1-10}$ on Lsg1 spGFP signal after heat shock. For estradiol-inducible systems that did not involve heat shock, populational mean spGFP intensity of each biological repeat was used for the following analyses. For the flow cytometry quantification, populational mean GFP intensities of at least 25,000 single cells were calculated for the following analyses. Most quantifications were shown as absolute intensity values with an arbitrary unit. Normalized spGFP intensities were calculated to highlight the relative changes between different strains.

## Mammalian cell line culture, transfection, imaging, and quantification

Human RPE-1 cells (ATCC CRL-4000, Manassas, VA, USA) were cultured in Dulbecco's Modified Eagle Medium: Nutrient Mixture F-12 (DMEM/F12) (Thermo Fisher Scientific), supplemented with 10% (vol/vol) fetal bovine serum, 100 IU/ml penicillin. Transient transfections were performed with Lipofectamine 3000 (Invitrogen) according to the manufacturer's instructions. The cell line has been authenticated by STR profiling (ATCC) and tested as mycoplasma negative.

RPE-1 cells were dually transfected with MTS-mCherry-$GFP_{1-10}$ and the protein of interest tagged with $GFP_{11}$ (2.5 µg of each plasmid was applied). For imaging, MatTek (P35G-0-14C) dish was used to culture cells, and cells were located using the mCherry channel only. Cells were imaged or analyzed by flow cytometry after 24 or 48 hr of transfection for $FUS^{P525L}$ or FlucDM, respectively. For flow cytometry analysis of $FUS^{P525L}$ spGFP system, cells were permeabilized with digitonin buffer (0.32 M sucrose, 5 mM $CaCl_2$, 3 mM Mg[acetate]$_2$, 0.1 mM EDTA, 10 mM Tris-HCl, 100 µg/ml digitonin) for 8–10 min, in order to remove spGFP signal outside of mitochondria in cytosol.

To evaluate cell death caused by $FUS^{P525L}$ overexpression, equal number of RPE-1 cells were seeded in six-well plates and transfected with GST or $FUS^{P525L}$, with or without AICAR. Compared to GST transfection control, $FUS^{P525L}$ resulted in significant floating dead cells. Number of attached cells after 24 hr of transfection were analyzed with Attune NxT flow cytometer as a proxy for cell viability.

## Cell lysates, immunoblots, and antibodies

For yeast experiments, 1–2 ml of yeast cells in the indicated background and medium was collected by centrifugation and snap-frozen in liquid nitrogen for storage. Pellets were disrupted, boiled in

120 µl 1× LDS sample buffer for 10 min, and vortexed with an equal volume of 0.5 mm acid-washed glass beads to break cells at 4°C for 2 min with a 1 min interval. Cell lysates were boiled for 5 min, separated from glass beads by 15,000 × g centrifugation at room temperature for 30 s, and analyzed by SDS-PAGE. For mammalian data, RPE-1 cells were washed with PBS and lysed with RIPA buffer (MilliporeSigma) supplemented with protease inhibitors on ice for 20–30 min. Cell lysates were further sonicated and incubated on ice for 5 min, followed by 10 min 21,200 × g centrifugation at 4°C. The supernatant was collected and analyzed by SDS-PAGE.

Transfer was performed using iBlot2 (Thermo Fisher Scientific) and immunoblots were developed using Clarity Western ECL substrate (Bio-Rad, Hercules, CA, USA) for HRP-linked secondary antibodies, or directly using fluorescent IRDye secondary antibodies (LI-COR, Lincoln, NE, USA). Images were acquired by using LI-COR imaging systems and analyzed in Image Studio (LI-COR). HA-tag (C29F4) rabbit mAb #3724 was purchased from Cell Signaling Technology (Danvers, MA, USA). PGK1 mouse mAb (22C5D8) was purchased from Invitrogen/Thermo Fisher Scientific. FLAG mouse clone M2 (F1804) was obtained from MilliporeSigma. GFP Living Colors A.v. mAb clone JL-8 (632381) was obtained from Takara Bio (Kusatsu, Shiga, Japan).

## Firefly luciferase assays

Firefly luciferase assays in yeast were carried out as described previously (*Nathan et al., 1997*). In brief, after 90 min of estradiol induction, 100 µl of cells was vigorously mixed with 100 µl of 1 mM D-luciferin in a white 96-well plate (655073, Greiner Bio-One, Kremsmünster, Austria), and light emission was immediately measured by the luminescence detection mode in Cytation 5 (Biotek, Winooski, VT, USA). Luciferase activities were normalized to cell density measured by $OD_{600}$ and adjusted to total abundance of FlucSM protein measured by immunoblotting.

## Mig1 nucleocytoplasmic translocation

The nucleocytoplasmic distribution of Mig1-GFP was quantified using a custom ImageJ macro and MATLAB script as described previously (*Kelley and Paschal, 2019*). In brief, nuclear protein Pus1-RFP was used to create nucleoplasmic mask for each cell (*Witkin et al., 2012*). Cytoplasm was defined by a dilated nuclear mask (*Kelley and Paschal, 2019*). The nuclear-cytoplasmic ratio of each cell was calculated by dividing the mean nuclear intensity by the mean cytoplasmic intensity. Populational mean nuclear-cytoplasmic ratio of at least three biological replicates were used for statistical analyses.

## Yeast growth curve

Yeast cells with indicated genetic background were cultured in corresponding media. Overnight cultures were refreshed for 4 hr at 30°C and the $OD_{600}$ of the cells was measured and adjusted to 0.05. Diluted cell suspension was added to a 96-well plate with 2 µM estradiol or ethanol as control. The wells along the perimeter of the plate were pre-filled with 200 µl cell-free medium to prevent evaporation. The $OD_{600}$ was continuously monitored at 30°C using Cytation 5 every 20 min with constant shaking. Data were extracted and analyzed using the R package GroFit (https://cran.r-project.org/src/contrib/Archive/grofit/) (*Kahm et al., 2010*).

## MMP measurements

Yeast cells expressing MPs and growing in appropriate medium was collected, incubated with 2.5 µM TMRM (21437, Cayman Chemical) for 15 min at 30°C and washed twice by fresh medium before recording with Attune NxT flow cytometer equipped with appropriate filter sets. A spGFP intensity threshold was applied so that less than 1% of cells displayed positive spGFP in the ethanol-treated control groups with no expression of MPs. Mean TMRM intensities of at least 25,000 cells were calculated for each biological replicate.

RPE-1 cells transfected with either GST or FUS[P525L] for 24 hr were washed once with PBS and added with complete media containing 150 nM TMRM for 30 min at 37°C. After incubation, cells were washed with PBS and trypsinized into single cells. Cell suspensions were pelleted and re-suspended in PBS for analysis on the Attune NxT flow cytometer.

## Tetrazolium overlay assay

Yeast tetrazolium overlay was performed to measure the respiratory deficiency in a yeast population as previously described (*Ogur et al., 1957*). In brief, yeast cells were inoculated in YPD media at 30°C overnight. Around 100 cells were plated on YPD plates and grew for 4 days at 30°C. The tetrazolium test medium consists of 1.5% agar and 0.1% tetrazolium (17342, Cayman Chemical) in 0.067 M phosphate buffer at pH 7.0. Test was performed by pouring 15 ml of melted test medium at 55°C over a YPD plate. The number of large red colonies (respiration-sufficient) and small white colonies (respiration-deficient) were counted after 1 hr of incubation at 30°C.

## Super-resolution imaging

Structured illumination microscopy (SIM) images were acquired with a GE OMX-SR Super-Resolution Microscope 3D Structure Illumination (3D-SIM) equipped with high-sensitivity PCO sCMOS cameras, or LSM880-Airyscan FAST Super-Resolution microscopy equipped with 63×/1.4 PlanApo oil. GFP and mCherry were excited with 488 and 568 nm lasers, respectively. The SIM images were reconstructed with the Softworx and aligned following the Applied Precision protocols, and Zeiss images were reconstructed with Airyscan processing. 3D rendering was performed with Imaris (Oxford Instruments Group).

## Statistical analysis

Descriptions of statistical tests and p values can be found in figure legends. At least three biological replicates (independent transformants) were analyzed in all experiments. Statistical analyses were performed with GraphPad Prism 6.0 and Microsoft Excel. No statistical methods were used to predetermine the sample size. No exclusion criteria were pre-established. The experiments were not randomized, and the investigators were not blinded to allocation during experiments and outcome assessment.

## Acknowledgements

We thank C Zhou and S Claypool for valuable discussion.

## Additional information

### Funding

| Funder | Grant reference number | Author |
|---|---|---|
| National Institutes of Health | Grant R35 GM118172 | Rong Li |
| ReStem Biotech | | Rong Li |
| American Heart Association | Predoctoral Fellowship AHA 17PRE33670517 | Linhao Ruan |
| Johns Hopkins University | Isaac Morris Hay and Lucille Elizabeth Hay Graduate Fellowship | Linhao Ruan |
| National Institutes of Health | BCMB graduate program at Johns Hopkins School of Medicine T32 GM007445 | Yuhao Wang Linhao Ruan Alexis Tomaszewski |

The funders had no role in study design, data collection and interpretation, or the decision to submit the work for publication.

### Author contributions

Yuhao Wang, Linhao Ruan, Conceptualization, Formal analysis, Investigation, Visualization, Methodology, Writing - original draft; Jin Zhu, Investigation, Methodology, Writing - original draft; Xi Zhang, Alexander Chih-Chieh Chang, Alexis Tomaszewski, Investigation, Methodology; Rong Li, Conceptualization, Supervision, Funding acquisition, Investigation, Methodology, Writing - review and editing

**Author ORCIDs**
Yuhao Wang http://orcid.org/0000-0002-2491-6916
Linhao Ruan http://orcid.org/0000-0002-6231-2566
Rong Li http://orcid.org/0000-0002-0540-6566

Joint public review: https://doi.org/10.7554/eLife.87518.3.sa1
Author response https://doi.org/10.7554/eLife.87518.3.sa2

## Additional files

**Supplementary files**
• MDAR checklist

**Data availability**

All data generated or analyzed during this study are included in the manuscript and supporting files. Source data files have been provided for all figures and figure supplements. More details about protocols, reagents, and newly created materials can be obtained from the corresponding author upon reasonable request.

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
