## [Editor Report · eLife assessment]

This study makes a connection between cellular metabolism and proteostasis through MAGIC, a previously proposed protein quality control pathway of clearance of cytosolic misfolded and aggregated proteins by importing into mitochondria. The authors reveal the role of Snf1, a yeast AMPK, in preventing the import of misfolded proteins to mitochondria for MAGIC controlled by the transcription factor Hap4, depending on the cellular metabolic status. The key message is **important**, although the evidence for physiological relevance of MAGIC for overall cellular proteostasis and its molecular regulation by Snf1 remains **incomplete**.

---

## [Referee Report · Joint public review]

In this study, Wang et al extend on their previous finding of a novel quality control pathway, the MAGIC pathway. This pathway allows misfolded cytosolic proteins to become imported into mitochondria and there they are degraded by the LON protease. Using a screen, they identify Snf1 as a player that regulates MAGIC. Snf1 inhibits mitochondrial protein import via the transcription factor Hap4 via an unknown pathway. This allows cells to adapt to metabolic changes, upon high glucose levels, misfolded proteins become imported and degraded, while during low glucose growth conditions, import of these proteins is prevented, and instead import of mitochondrial proteins is preferred.

This is a nice and well-structured manuscript reporting on important findings about a regulatory mechanism of a quality control pathway. The findings are obtained by a combination of mostly fluorescent protein-based assays. Findings from these assays support the claims well.

While this study convincingly describes the mechanisms of a mitochondria-associated import pathway using mainly model substrates, my major concern is that the physiological relevance of this pathway remains unclear: what are endogenous substrates of the pathway, to which extent are they imported and degraded, i.e. how much does MAGIC contribute to overall misfolded protein removal (none of the experiments reports quantitative "flux" information). Lastly, it remains unclear by which mechanism Snf1 impacts on MAGIC or whether it is "only" about being outcompeted by mitochondrial precursors.

---

## [Author Response]

The following is the authors’ response to the original reviews.

**Reviewer #1 (Public Review):**
"MAGIC" was introduced by the Rong Li lab in a Nature letters article in 2017. This manuscript is an extension of this original work and uses a genome wide screen the Baker's yeast to decipher which cellular pathways influence MAGIC. Overall, this manuscript is a logical extension of the 2017 study, however the manuscript is challenging to follow, complicated by the data often being discussed out of sequence. Although the manuscripts make claims of a mechanism being pinpointed, there are many gaps and the true mechanisms of how the factors identified in the screen influence MAGIC is not clear. A key issue is that there are many assumptions drawn on previous literature, but central aspects of the mechanisms being proposed are not adequately shown.Key comments:1. Reasoning and pipelines presented in the first two sections of the results are disordered and do not follow figure order. In some instances, the background to experimental analyses such as detailing the generation of spGFP constructs in the YKO mutant library, or validation of Snf1 activation are mentioned after respective results are discussed. This needs to be fixed.

We thank the reviewer for pointing out potential confusion to readers. We have revised the first two sections according to reviewer’s suggestion. (Page 4-6)

2. In general there is a lack of data to support microscopy data and supporting quantification analysis. The validity of this data could be significantly strengthened with accompanying western blots showing accumulation of a given constructs in mitochondrial sub compartments (as was the case in the lab’s original paper in 2017).

We appreciate the reviewer’s suggestion on biochemical validations. However, the validity of this imaging-based assay for detecting import of cytosolic misfolded proteins into mitochondria, including the use of FlucSM as a model misfolding-prone protein, was carefully established in our previous study by using appropriate controls, super resolution imaging, APEX-based proximity labeling, and classical biochemical fractionation and protease protection assay (Ruan et al., 2017 Nature, ref. 10). We have reminded readers of these validation experiments in the previous study on Page 4, line 14-17.

In recent years, advancements in imaging-based tools have allowed many protein interactions and dynamic processes, which were previously examined by using biochemical assays in lysates of populations of cells, to be observed with various level of quantitation in live cells with intact cellular compartments. Many of these assays, e.g., the RUSH assay for ER to Golgi transport, FRAP-based analysis for nuclear/cytoplasmic shuttling of proteins, or FRET-based assays for protein-protein interactions, have been well accepted and even embraced by the respective fields of study once validated with genetic and biochemical approaches. The advantages for live-cell imaging-based assays are often their unique ability to report dynamic processes or unstable molecular species with spatiotemporal sensitivity. Respectfully, it is our view, based on our own experience, that the traditional protease protection assay is not adequate or sufficiently quantitative for examining the presence of unstable misfolded proteins in mitochondrial sub-compartments, given the obligatorily lengthy in vitro cell lysis and mitochondrial isolation process, during which the unstable proteins are continuously being degraded. This likely explains our previous biochemical fractionation result that only weak protein signals were detected in the matrix fraction (Ruan et al., 2017 Nature, ref. 10). In addition, unlike stably folded, native mitochondrial matrix proteins, misfolded/unfolded proteins such as Lsg1 or FlucSM are highly susceptible to protease treatment. This sensitivity makes the assay unreliable for detecting such proteins if trace amount of the protease penetrates mitochondrial membranes during cell lysis even without detergent treatment.

While we agree that protease protection assay is highly valuable for qualitative detection of the presence of a protein in certain mitochondrial compartments or determining its topology on membranes, this assay (regrettably in our hands) does not allow quantitative comparisons that were necessary for this study, because of inherent sample to sample variation, yet the laborious and low throughput nature of this assay makes it difficult for adequate statistical analysis. Furthermore, the level of protein detection in various fractions is highly sensitive to how the sample is treated with protease and detergent. Our imaging-based quantification, on the other hand, allows us to compare increased or decreased presence of GFP11-tagged proteins in mitochondria under different metabolic conditions or in different mutant or wild-type strains. Data from hundreds of cells and at least three independent biological replicates allowed us to apply adequate statistical analysis to aid our conclusion.

3. Much of the mechanisms proposed relies on the Snf1 activation. This is however not shown but assumed to be taking place. Given that this activation is central to the mechanism proposed, this should be explicitly shown here - for example survey the phosphorylation status of the protein.

Both REG1 deletion and low glucose conditions have been demonstrated extensively for Snf1 phosphorylation and activation in yeast (e.g., many seminal papers from Marian Carlson’s and other lab, such as ref. 24-28). In our study, we have indeed corroborated this by showing that Mig1 was exported from the nucleus in Δreg1 mutant and in low glucose conditions (Figure 1—figure supplement 2H and I). The mechanism of Snf1-mediated nuclear export of Mig1 has been characterized in detail as well (e.g., ref. 29-31).

Recommendations for the authors: please note that you control which, if any, revisions, to undertake

**Reviewer #1 (Recommendations For The Authors):**
SPECIFIC COMMENTSGenetic Screen o Line 20 - the narrative moves to SNF1, but the reasoning for the selection of this Class I substrate is not defined. What was the basis for this selection - what happened to the other Class I substrates. It is stated in the text that the other Class I proteins show the same increase in spGFP signal. The data showing this should be included in the Supp Figure 1 for transparency.

We have moved the narratives of Snf1 function to the second section and clarified that we were interested in this gene due to its central role in metabolism and mitochondrial functions that may influence MAGIC (Page 5: line 16-20). Other genes in class 1 were shown in Table S1. Detailed discussion of other genes in this category is beyond the scope of this study.

Snf1/AMPK prevents MP accumulation in mitochondria:The FlucDM data in human RPE-1 mitochondria seems to be added to only increase the significance of the work. The mechanisms suggested here with Hap4 would not be possible in human cells as there is no homologue of this protein in human cells. Making generalisations that these pathways are conserved based on this one experiment is not appropriate.

We appreciate this feedback. Although the focus of this study is the regulation of MAGIC by the yeast AMPK Snf1, we would like to share our initial observation that suggests a similar role of AMPK in human RPE-1 cells. We acknowledge that the underlying mechanisms regarding the downstream transcription factors and pathway for misfolded protein import could be different in mammalian cells, but the overall effect of AMPK in mitochondrial biogenesis is well known to resemble that of Snf1. To avoid making over-generalization, we changed our statement of conclusion to: ‘These results suggest that AMPK in human cells regulates MP accumulation in mitochondria following a similar trend as in yeast, although the underlying mechanisms might differ between these organisms.’ (Page 7: line 2-4)

Mechanisms of MAGIC regulation by Snf1:While the lysosome is ruled out here the authors have not considered the proteasomes. Is there a reason for this? Given accumulation of aggregates outside of mitochondria, and previous connections of the proteasome to mitochondrial quality control this would be an obvious thing to check.We examined the role of lysosomal degradation here because it is known to be activated under Snf1active condition (ref. 37). We appreciate this feedback and have included a new analysis on MG132treated FlucSM spGFP strains in which PDR5 gene was deleted to avoid drug efflux.

This result suggests that the proteosome inhibitor did not ablate the difference in FlucSM accumulation between these conditions. That MG132 promoted mitochondrial accumulation of FlucSM in both high glucose and low glucose conditions was not surprising, as FlucSM is also degraded by proteasome in the cytosol (Ruan et al., 2017 Nature, ref. 10), and preventing this pathway could divert more of such protein molecules toward MAGIC. (Page 7: line 26-29).

Line 13 "we hypothesized that elevated expression of mitochondrial preproteins induced by the activation of Snf1-Hap4 axis (REF) may outcompete MPs for import channels". This statement has some assumptions. The authors have not shown that Snf1 is activated in thier models and more importantly that they have an accumulation of mitochondrial preproteins. The data that follows using the cytosolic domains of the receptors is hard to rationalise without seeing evidence that there is in fact pre-protein accumulation or impacts on the mitochondrial proteome in this system.

As stated in our response to main point [3], Snf1 activation in reg1 mutant or in low glucose is evidenced by our data showing Mig1 export from nucleus to cytoplasm and had also been shown in many previous publications. A recent study (Tsuboi et al., 2020 eLife) also showed a dramatic increase in mitochondrial volume fraction in Δreg1 cells and wild-type cells in respiratory conditions, further supporting the role of Snf1 in mitochondrial biogenesis. We have provided relevant references in the manuscript (ref. 24-28).

The ability of Tom70 cytosolic domain (Tom70cd), which can bind mitochondrial preproteins but not localize to mitochondria due to lack of N-terminal targeting sequence, to compete with endogenous Tom70 for mitochondrial preproteins has been well documented (ref. 47-49). However, we agree with the reviewer that a future quantitative proteomics study to measure changes in mitochondrial proteome under Tom70cd over-expression could allow more accurate interpretation of our experimental result.

AMPK protects cellular fitness during proteotoxic stress:The inhibition of preprotein import by overexpressing the cytosolic domains of receptors is not supported with some proof of principle data. If this was working as the authors assume, it is not clear why only an effect with Tom70 is observed. The majority of the mitochondrial proteome is imported via Tom20/Tom22 so this does not align with what the authors are suggesting. Is the Tom70CD and any associated Hsp proteins facilitating the observed changes to the MPs?

We thank the reviewer for raising this point. We expressed different TOM receptor cytosolic domains but found that Tom70cd had the strongest rescue on MAGIC under AMPK activation conditions. It is possible that certain Tom70 substrates or Tom70-assoicated heat shock proteins inhibit the import of MAGIC substrates. We admit that a clear explanation of this unexpected observation necessitates a better understanding of how native and MAGIC substrates are selected and imported by the outer-membrane channel. We can only offer our best interpretation based on the current state of the understanding, and we feel that we have been careful to acknowledge such in the manuscript.

While the effect of AMPK inactivation reducing FUS accumulation was striking, this was all in the context of overexpression and may not be physiologically relevant - or may occur very transiently under basal conditions. Is GST an appropriate control here, why not use WT FUS? Likewise, one representative image is shown in Figure 5 - can the authors show western blotting that mitochondrial accumulation of FUS can be reduced with AMPK activation?

We thank the reviewer for this suggestion, however, overexpressed FUS WT is also aggregation prone(Zhihui Sun et al., 2011, PloS Biology; Shulin Ju, 2011, PloS Biology; Jacqueline C. Mitchell et., 2013,Acta Neuro). We believe that GST, as a well-folded protein, is an appropriate control (Ruan et al., 2017 Nature, ref. 10). As we discussed in response to main point [1], the in vitro assay involving protease protection and western blots do not allow reliable quantitative comparison in our hands.

In text changes.The analysis pipeline of the YKO mutant library should be introduced at the very start of the first paragraph, not the end.

Addressed on Page 4, second paragraph

"Fluc" should be introduced as "Firefly luciferase" within the first paragraph of the first section, also need to define SM and DM in FlucSM/FlucDM - these appear to be missing.

Addressed in both Introduction (Page 2: line 29; Page 3: line 8-9) and re-clarified in Result (Page 5: line27-29)

The role of Reg1 should be explicitly stated in the text, not just in the figure.

Addressed on Page 6: line 3-6

Figure 1H legend states Reg1 (WT) is Snf1-inactive and Reg1 KO is Snf1-active. This wording is confusing and is not supported by data, but by assumption. If the authors want to use this wording then evidence needs to be provided - as suggested above.

We have changed this and other legends to only show genotypes and medium conditions.

"Tom70cd overexpression also exacerbated growth rate reduction due to FlucSM expression in HG medium (Figure 4A; Figure 4 - figure supplement 1A)" should be figure supplement 1B.

Fixed on Page 10: line 10

"These results suggest that glucose limitation protects mitochondria and cellular fitness during FlucSM induced proteotoxic stress through Snf1-dependent inhibition of MP import into mitochondria". The phrase "Snf1-dependent inhibition of MP import into mitochondria" may be misleading, as Snf1 isn't modulating import directly but is acting on transcriptional regulators to modulate mitochondrial import under stress.

We restated the conclusion as follows: ‘These results suggest that Snf1 activation under glucose limitation protects mitochondrial and cellular fitness under FlucSM-associated proteotoxic stress.’ (Page 10: line 20-21).

"... Significantly increased the fraction of spGFP-positive and MMP-low cells in both HG and LG medium (Figure 4G-K)" should be (Figure 4J-K).

Fixed on Page 11: line 3.

**Reviewer #2 (Public Review):**
Work of Rong Li´s lab, published in Nature 2017 (Ruan et al, 2017), led the authors to suggest that the mitochondrial protein import machinery removes misfolded/aggregated proteins from the cytosol and transports them to the mitochondrial matrix, where they are degraded by Pim1, the yeast Lon protease. The process was named mitochondria as guardian in cytosol (MAGIC).The mechanism by which MAGIC selects proteins lacking mitochondrial targeting information, and the mechanism which allows misfolded proteins to cross the mitochondrial membranes remained, however, enigmatic. Up to my knowledge, additional support of MAGIC has not been published. Due to that, MAGIC is briefly mentioned in relevant reviews (it is a very interesting possibility!), however, the process is mentioned as a "proposal" (Andreasson et al, 2019) or is referred to require "further investigation to define its relevance for cellular protein homeostasis (proteostasis)" (Pfanner et al, 2019).Rong Li´s lab now presents a follow-up story. As in the original Nature paper, the major findings are based on in vivo localization studies in yeast. The authors employ an aggregation prone, artificial luciferase construct (FlucSM), in a classical split-GFP assay: GFP1-10 is targeted to the matrix of mitochondria by fusion with the mitochondrial protein Grx5, while GFP11 is fused to FlucSM, lacking mitochondrial targeting information. In addition the authors perform a genetic screen, based on a similar assay, however, using the cytosolic misfolding-prone protein Lsg1 as a read-out.My major concern about the manuscript is that it does not provide additional information which helps to understand how specifically aggregated cytosolic proteins, lacking a mitochondrial targeting signal could be imported into mitochondria. As it stands, I am not convinced that the observed FlucSM-/Lsg1-GFP signals presented in this study originate from FlucSM-/Lsg1-GFP localized inside of the mitochondrial matrix. The conclusions drawn by the authors in the current manuscript, however, rely on this single approach.In the 2017 paper the authors state: "... we speculate that protein aggregates engaged with mitochondria via interaction with import receptors such as Tom70, leading to import of aggregate proteins followed by degradation by mitochondrial proteases such as Pim1." Based on the new data shown in this manuscript the authors now conclude "that MP (misfolded protein) import does not use Tom70/Tom71 as obligatory receptors." The new data presented do not provide a conclusive alternative. More experiments are required to draw a conclusion.In my view: to confirm that MAGIC does indeed result in import of aggregated cytosolic proteins into the mitochondrial matrix, a second, independent approach is needed. My suggestion is to isolate mitochondria from a strain expressing FlucSM-GFP and perform protease protection assays, which are well established to demonstrate matrix localization of mitochondrial proteins. In case the authors are not equipped to do these experiments I feel that a collaboration with one of the excellent mitochondrial labs in the US might help the MAGIC pathway to become established.

We thank Reviewer 2 for these suggestions, but we would like to respectfully offer our difference in opinion:

a. Regarding the suggestion “to isolate mitochondria from a strain expressing FlucSM-GFP and perform protease protection assays”, in our previous study (Ruan et al., 2017 Nature, ref. 10), we have indeed applied two independent biochemical approaches: APEX-mitochondrial matrix proximity labeling and classic protease protection assay using non-spGFP strains, both consistently confirmed the entry of misfolded proteins into mitochondria under proteotoxic stress. Our super-resolution imaging further confirmed the import of the split GFP-labeled proteins to be inside mitochondria. Moreover, as we discussed in response to Reviewer 1’s main point [2], while the suggested biochemical assay is useful for validating topology within mitochondria, it is not quantitative and may not reliably report the in vivo accumulation of misfolded proteins in mitochondria due to the isolation process that takes hours, during which the unstable proteins could be continuously degraded within mitochondria.

While we agree with the reviewer that we do not yet understand how misfolded proteins are imported into mitochondria, it would be unfair to state “as it stands, I am not convinced..” simply because the underlying mechanism remains to be elucidated. We would like to point out that targeting sequences for many well-established mitochondrial proteins are still not well defined. It is well known that mitochondrial targeting sequences are not as uniformly predictable as, for example, nuclear targeting sequences. Our finding that deletion of TOM6 enhances the import of misfolded proteins suggest that their import may involve the TOM channel in a more promiscuous conformation, which may reduce the requirement for a specific sequence-based targeting signal associated with the substrate.

b. Regarding the role of Tom70, in our 2017 study, using proteomics and subsequently immunoprecipitation we validated the binding, albeit not necessarily direct, between misfolded protein FlucSM and Tom70. Therefore, “we speculate that protein aggregates engaged with mitochondria via interaction with import receptors such as Tom70”. Recent studies from different labs confirmed the interactions between Tom70 and aggregation prone proteins (Backes et al., 2021, Cell Reports; Liu et al., 2023, PNAS). In the current study, surprisingly, knockout of TOM70 did not block MAGIC, suggesting redundant components of mitochondria import system may facilitate the recruitment of misfolded proteins in the absence of Tom70, and this does not contradict the notion that Tom70 helps tether protein aggregates to mitochondria.

c. Regarding other studies also showing the import of misfolding or aggregation-prone cytosolic proteins into mitochondria, there have been at least several recent studies in the literature for mammalian cells involving either model substrates or disease proteins (e.g., ref. 12-15; 56-58; Vicario, M. et al. 2019 Cell Death Dis.). The studies are briefly mentioned in Introduction (Page 3, paragraph 2). The present manuscript documents a major effort from our group using whole genome screen in yeast to understand the mechanism and regulation of MAGIC. Many of the screen hits have yet to be studied in detail. We full agree that much remains to be understood about whether and how this pathway affects proteostasis and what might be the evolutionary origin for such a mechanism.

Additional comments:The genetic screen:The genetic screen identified five class 1 deletion strains, which lead to enhanced accumulation of Lsg1GFP and a larger set of class 2 mutants, which lead to reduced accumulation. Please note, in my opinion it is not clear that accumulation of the reporters occurs inside the mitochondria. In any case, the authors selected one single protein for further analysis: Snf1, the catalytic subunit of the yeast SNF complex, which is required for respiratory growth of yeast.The results of the screen are not discussed in any detail. The authors mention that ribosome biogenesis factors are abundant among class 2 mutants. Noteworthy, Lsg1 is involved in 60S ribosomal subunit biogenesis. As Lsg1-GFP11 is overexpressed in the screen this should be discussed. Class 2 mutants also .include several 40S ribosomal subunit proteins (only one of the 60S subunit). What does this imply for theMAGIC model? Also, it should be discussed that the screen did not identify Δreg1 and Δhap4, which I had expected as hits based on the data shown in later parts of the manuscript.

We apologize for the confusion, but the GFP11 tag was in fact knocked into the C-terminus of Lsg1 in the endogenous LSG1 locus, and so Lsg1 was not overexpressed in the screen. We have made sure that this information is clearly conveyed in the revised manuscript (Page 4: line 20-22). How the ribosome small subunit affects MAGIC is beyond the focus of the current study and will be pursued in the future.

Regarding why certain mutants did not come out of our initial screen, this is not unexpected as the YKO collection, although extremely valuable to the community, is known to be potentially affected by false knockouts, suppressor accumulation and cross contamination (for references, e.g., Puddu et al., 2019 Nature). Additionally, high-through screens can also miss real hits. In our experience using this collection in several studies, we often found additional hits from analysis of genes implicated by known genetic or biochemical interactions.

Mutant yeast strains and growth assays:The Δreg1 strain grows poorly in all growth conditions and frequently accumulates extragenic suppressor mutations (Barrett et al, 2012). It would be good to make sure that this is not the case in the strains employed in this study. My suggestion is to do (and show) standard yeast plating assays with the relevant mutant strains including Δreg1, snf1, hap4, Δreg1Δhap4 without the split GFP constructs and also with them (i.e. the strains that were used in the assays).

We thank the reviewer for the suggestion. We were indeed aware of potential accumulation of suppressor mutations from the YKO library. Therefore, deletion mutants like Δreg1 and loss of TFs downstream of Snf1 that we used in the study after the initial screen were all freshly made and validated. At least 3 independent colonies were analyzed for each mutant (mentioned in Methods & Materials; Page 33, line 57). Moreover, the plating assay suggested here may not reveal additional information other than growth, which was taken into consideration during our experiments.

Activation of Snf1 in the relevant strains should be tested with the commercially available antibody recognizing active Snf1, which is phosphorylated at Snf1-T210.

Snf1 activation was validated by the Mig1 exporting from the nucleus. We also noted above that many studies have clearly demonstrated Snf1 activation in reg1 mutant and under low glucose growth (e.g., ref. 24-28).

Effects of Snf1, Reg1, Hap4 and respiratory growth conditions:The authors show that split GFP reporters show enhanced accumulation during fermentative growth, in Δsnf1, and Δreg1Δhap4 and fail to accumulate during respiratory growth, in Δreg1 and upon overexpression of HAP4. Analysis of Δhap4 should be included in Fig. 2. The suggestion that upon activation of Snf1 enhanced Hap4-dependent expression "outcompetes" misfolded protein import seems unlikely as only a fraction of mitochondrial genes is under control of Hap4. Without further experimental evidence I do not find that a valid assumption. More likely, the membrane potential plays a role: it is low during fermentative growth, in Δsnf1 and Δreg1Δhap4, and high during respiratory growth and in Δreg1 (Hübscher et al, 2016). Such an effect of the membrane potential seems to contradict the findings in the 2017 paper and the issue should be clarified and discussed. In any case, these data do not reveal that GFP reporters accumulate inside of the mitochondria. Based on the currently available evidence they may accumulate in close proximity/attached to the mitochondria. This has to be tested directly (see above).

We have included our analysis of Δhap4 in Page 8: line 14-15 and Figure 2—figure supplement 1H. Consistent with our result for Δreg1Δhap4 in glucose-rich medium, HAP4 deletion also resulted in a significant increase in mitochondrial accumulation of FlucSM in low glucose medium compared to WT. It did not have effect in high glucose condition in which Snf1 is largely inactive.

It is our view that the importance of Hap4 should not be judged by the number of nuclear encoded mitochondrial proteins they regulate. Still, this sub-group comprises a considerable number of proteins (at least 55 genes upregulated by Hap4 overexpression, ref. 43), and certain substrates may be more competitive with misfolded cytosolic proteins for import. Our genetic data strongly suggest that the inhibitory effect of active Snf1 on MAGIC is through Hap4, although we agree with the reviewer that detailed mechanism on how Hap4 substrates may compete with misfolded proteins need to be addressed in future studies.

Membrane potential is important for mitochondrial import. During respiratory growth and in Δreg1, membrane potential is well known to be elevated comparing to fermentative condition (e.g., Figure 4C). Our observation that the import of misfolded proteins into mitochondria is reduced under these conditions simply suggests that this reduction is not due to a lack of membrane potential. This is not in any way contradictory to our 2017 finding that misfolded protein import requires membrane potential (ref. 10).

Again, the accumulation of misfolded proteins in mitochondria, especially the model protein FlucSM, has been validated by using super resolution imaging (Figure 1—figure supplement 1A) in addition to the protease protection assay in our 2017 study.

Introduction and Discussion:Both are really short, too short in my view. Please provide some background of the general principals of mitochondrial protein import and information of how exactly translocation of cytosolic, aggregated proteins (lacking targeting information) is supposed to work. I do not understand exactly how the authors actually envisage the process.

We thank the reviewer for the suggestion. In the revised manuscript, we have extended both Introduction (Page 2-3) and Discussion section (Page 11-13)

The results from the 2022 eLife paper (Liu et al, 2022), which suggests that Tom70 may "regulate both the transcription/biogenesis and import of mitochondrial proteins so the nascent mitochondrial proteins do not compromise cytosolic proteostasis or cause cytosolic protein aggregation" should be discussed with regard to the data obtained with overexpression of the Tom70 soluble domain.

We thank the reviewer for pointing out that study and we have included a brief comment in Discussion section (Page 12: line 13-16). As the function of Tom70 appears to be complex, we cannot exclude the possibility that overexpression of the cytosolic domain has additional or indirect effects in addition to that due to preprotein binding.

Andreasson, C., Ott, M., and Buttner, S. (2019). Mitochondria orchestrate proteostatic and metabolic stress responses. EMBO Rep 20, e47865.Barrett, L., Orlova, M., Maziarz, M., and Kuchin, S. (2012). Protein kinase A contributes to the negative control of Snf1 protein kinase in *Saccharomyces cerevisiae*. Eukaryot Cell 11, 119-128.Hubscher, V., Mudholkar, K., Chiabudini, M., Fitzke, E., Wolfle, T., Pfeifer, D., Drepper, F., Warscheid, B., and Rospert, S. (2016). The Hsp70 homolog Ssb and the 14-3-3 protein Bmh1 jointly regulate transcription of glucose repressed genes in *Saccharomyces cerevisiae*. Nucleic Acids Res. 44, 5629-5645.Liu, Q., Chang, C.E., Wooldredge, A.C., Fong, B., Kennedy, B.K., and Zhou, C. (2022). Tom70-based transcriptional regulation of mitochondrial biogenesis and aging. Elife 11Pfanner, N., Warscheid, B., and Wiedemann, N. (2019). Mitochondrial proteins: from biogenesis to functional networks. Nat Rev Mol Cell Biol 20, 267-284.Ruan, L., Zhou, C., Jin, E., Kucharavy, A., Zhang, Y., Wen, Z., Florens, L., and Li, R. (2017). Cytosolic proteostasis through importing of misfolded proteins into mitochondria. Nature 543, 443-446.I prefer to have "all in one", also due to time limitation.It would be great to be able to upload the review file as otherwise formatting and symbols get lost.
**Reviewer #3 (Public Review):**
In this study, Wang et al extend on their previous finding of a novel quality control pathway, the MAGIC pathway. This pathway allows misfolded cytosolic proteins to become imported into mitochondria and there they are degraded by the LON protease. Using a screen, they identify Snf1 as a player that regulates MAGIC. Snf1 inhibits mitochondrial protein import via the transcription factor Hap4 via an unknown pathway. This allows cells to adapt to metabolic changes, upon high glucose levels, misfolded proteins an become imported and degraded, while during low glucose growth conditions, import of these proteins is prevented, and instead import of mitochondrial proteins is preferred.This is a nice and well-structured manuscript reporting on important findings about a regulatory mechanism of a quality control pathway. The findings are obtained by a combination of mostly fluorescent protein-based assays. Findings from these assays support the claims well.While this study convincingly describes the mechanisms of a mitochondria-associated import pathway using mainly model substrates, my major concern is that the physiological relevance of this pathway remains unclear: what are endogenous substrates of the pathway, to which extend are they imported and degraded, i.e. how much does MAGIC contribute to overall misfolded protein removal (none of the experiments reports quantitative "flux" information). Lastly, it remains unclear by which mechanism Snf1 impacts on MAGIC or whether it is "only" about being outcompeted by mitochondrial precursors.

We thank Reviewer 3 for the positive and encouraging comments on our manuscript. We agree with the reviewer that identifying MAGIC endogenous substrates and understanding what percentage of them are degraded in mitochondria are very important issues to be addressed. We are indeed carrying out projects to address these questions. We also agree with Reviewer 3 that the effect of Snf1 on MAGIC may have additional mechanisms in addition to precursors competition, such as Tom6 mediated conformational changes of TOM pores. In the revised manuscript, we had added a discussion to address these comments (Page 12: line 21-28).

**Reviewer #3 (Recommendations For The Authors):**
1. In their screen, the authors utilize differences in GFP intensity as a measure for import efficiency. However, reconstitution of the GFP from GFP1-10 and GFP11 in the matrix might also be affected (folding factors, differential degradation).

Upon Snf1 activation, the protein abundance of mitochondrial chaperones such as Hsp10, Hsp60, and Mdj1, and mitochondrial proteases such as Pim1 are not significantly changed (ref. 35). Therefore, it is unlikely that the folding and degradation capacity of mitochondrial matrix is drastically affected by Snf1 activation.

To examine the effect of Snf1 activation on spGFP reconstitution, Grx5 spGFP strain was constructed in which the endogenous mitochondrial matrix protein Grx5 was C-terminally tagged with GFP11 at its genomic locus, and GFP1-10 was targeted to mitochondria through cleavable Su9 MTS (MTS-mCherryGFP1-10) (ref. 10). Only modest reduction in Grx5 spGFP intensity was observed in LG compared to HG, and no significant difference after adjusting the GFP1-10 abundance (spGFP/mCherry ratio) (Figure 1— figure supplement 3A-D). These data suggest that any effect on spGFP reconstitution is insufficient to explain the drastic reduction of MP accumulation in mitochondria under Snf1 activation. Overall, our results demonstrate that Snf1 activation primarily prevents mitochondrial accumulation of MPs, but not that of normal mitochondrial proteins. (Page 6: line 17-25).

We admit, however, that to fully rule out these factors, specific intra-mitochondrial folding or degradation reporter assays would be needed.

1. Scoring of protein import always takes place using fluorescence-based assays. These always require folding of the "sensors" in the matrix. An additional convincing approach that would not rely on matrix folding could be pulse chase approaches coupled to fractionation assays and immunoprecipitation.

We thank reviewer 3 for this suggestion. In our previous study, we applied two different biochemical assays: APEX proximity labeling, and mitochondrial fractionation followed by protease protection. Both confirmed the entry of misfolded proteins into mitochondria as observed by using split GFP. As we discussed in response to Reviewer 1’s main point [3], the fractionation assays are not quantitative enough for the comparisons made in our study. In particular, during the over 2-hour assay, misfolded proteins continue to be degraded within mitochondria. By using proper controls, our spGFP system provides quantitative comparisons for mitochondrial accumulation of misfolded proteins in non-disturbed physiological conditions.

2. Could the pathway be reconstituted in vitro with isolated mitochondria to test for the "competition hypothesis"

This is an excellent suggestion, but setting up such a reconstituted system is a project on its own. The study documented in this manuscript already encompasses a large amount of work that we feel should be published timely.

3. Fluorescence figures are not colour blind friendly (red-green). This should be improved by changing the color scheme.

We thank reviewer 3 for pointing this out and sincerely apologize for any inconvenience. However, we are unfortunately unable to change all images within a limited time. We will adopt another color scheme in future work.

4. spGFP in human cells appears to form "spot-like" structures. What are these granules?

We indeed observed granule-like structures by spGFP labeled FUS in mitochondria, which is interesting, but we did not investigate this further because it is a not a focus of this study.